# Fully chemical interface engineering for statically and dynamically stable perovskite solar cells

Luyao Li[1,2], Cheng Wang[1], Weicun Chu[1], Jaewang Park[3], Yiming Dai[1], Qiankai Ba[4], Kaifeng Wang[4], Jiaxing Gao[1], Zeliang Wei[1], Xiaoming Zhao[1], Xuchen Nie[1], Lixiong Yin[2]✉, Sang Il Seok[3]✉, Riming Nie[1]✉ & Wanlin Guo[1]

The interfacial modifications between perovskite and charge-transport layers can arise from strong chemisorption bonds or weak physical adsorption interactions. However, modifications based on physical adsorption are susceptible to detachment, which not only disrupts the original energy level alignment and defect passivation but also introduces new charge recombination centers. Here, we report a fully chemical modification strategy in which the interfacial modifiers undergo an in situ crosslinking-like reaction, forming a localized, chemically bonded layer that seamlessly extends from the bulk of the underlying transport layer to the interface. Perovskite solar cells (PSCs) fabricated with this fully chemical modification strategy achieve a power conversion efficiency (PCE) of 25.52% (certified 25.49%) under standard conditions, representing one of the highest PCEs reported for devices fully fabricated in an ambient atmosphere. In terms of static stability, unencapsulated devices exhibit linear extrapolated $T_{80}$ lifetimes of 27,000 h during dark shelf storage and 19,000 h under thermal stress at 85 °C, both of which are record-breaking values for dark shelf and thermal stability, respectively. For dynamic stability, the devices maintain a linear extrapolated $T_{80}$ lifetime of 2,600 h under light-dark cycling, representing the most dynamically stable PSCs reported to date.

Perovskite solar cells (PSCs) are widely recognized as one of the most promising next-generation photovoltaic materials, with the power conversion efficiency (PCE) of single-junction PSCs now exceeding 26%[1–4]. However, a crucial challenge lies in enhancing their durability to achieve a service life comparable to silicon-based cells. The degradation of PSCs is typically triggered by defects at the interfaces or grain boundaries, which not only lead to non-radiative charge recombination losses but also severely compromise the stability of PSCs[4,5].

Interface engineering between the perovskite absorption layer and the charge transport layers is crucial for enhancing the efficiency and stability of PSCs. The widespread interest in interface modification can be attributed to the following reasons: (i) the presence of defects in the various functional layers (perovskite absorption layer, electron transport layer, and hole transport layer), which requires passivations; (ii) the incomplete alignment of energy levels between these functional layers, necessitating energy level tuning; and (iii) the potential

[1]State Key Laboratory of Mechanics and Control of Mechanical Structures, Key Laboratory for Intelligent Nano Materials and Devices of the Ministry of Education, Institute for Frontier Science, Nanjing University of Aeronautics and Astronautics, Nanjing, PR China. [2]School of Materials Science and Engineering, Shaanxi University of Science and Technology, Xi'an, PR China. [3]Department of Energy Engineering, School of Energy and Chemical Engineering, Ulsan National Institute of Science and Technology, Ulju-gun, Ulsan, Republic of Korea. [4]Shanghai Shengjian Technology Co., Ltd., Shanghai, China. ✉e-mail: ylx@sust.edu.cn; seoksi@unist.ac.kr; rmnie@nuaa.edu.cn

chemical reactions between metal oxides and perovskites, which adversely affect the efficiency and stability of PSCs. Numerous passivation strategies have been reported, such as inorganic salts, organic salts, ionic liquids, and self-assembled monolayers[6–13], aiming to repair interfacial defect states and improve the durability of the devices.

However, modifiers at the interface can be either strongly bonded through chemisorption or weakly bonded through physisorption, with the latter often becoming unstable after prolonged device operation[14,15]. During the device fabrication process, modifiers with high solubility are highly susceptible to being dissolved and washed away by solvents such as N, N′-dimethylformamide (DMF) in practical applications, which leads to the inability to maintain the passivation effects that were initially intended, ultimately failing passivation[16]. Furthermore, during the storage and operation of the device, the relatively weak binding energy can easily lead to the detachment of modifiers. The occurrence of such detachment can further generate new interface defects, which are undoubtedly highly detrimental to the device's performance[16,17]. In addition, environmental stress factors, such as humidity, heat, and light, can gradually form traps at the bottom of the perovskite active layer[18,19]. The formation of these traps can lead to severe ion migration at the interface, thereby greatly affecting the stability of the device[20–23]. Developing failure-resistant interface modification methods is crucial for enhancing device stability under both static conditions (constant environmental parameters such as fixed temperature, humidity, or continuous illumination) and dynamic conditions (simulating real-world scenarios including thermal cycling, light-dark cycling, or actual outdoor exposure).

Here, we first confirmed the detachment of interface modification materials (potassium chloride, guanidine hydrochloride, and methylammonium acetate) commonly used in normal (n-i-p) devices through theoretical calculations and experimental validation. Then, we take the typical $SnO_2$ electrode in the n-i-p structure as an example and propose a comprehensive chemical adsorption strategy. Specifically, we first pre-embed diethylenetriaminepentaacetic acid (**DTPA**) molecules within the $SnO_2$ layer, and then introduce zolephonic acid (**Zol**) on its surface to promote an in situ cross-linking-like reaction between the two modifiers, thereby establishing a local full chemical adsorption layer from the interior of the $SnO_2$ layer to the bottom of the PVK layer. The core purpose of this method is to significantly enhance the extraction and migration capabilities of interface charges and ensure that the modifiers do not fail. Ultimately, the PCE was significantly improved to 25.52% (certified 25.49%), one of the highest PCEs for devices fully fabricated in an ambient atmosphere. In addition, the target device showed excellent stability in multiple tests, such as dark storage, thermal stability, light-dark cycling tests, and maximum power point (MPP) tracking. It is worth mentioning that these unencapsulated devices exhibited linear extrapolated $T_{80}$ lifetimes of 27,000 h during dark shelf storage and 19,000 h under thermal stress at 85 °C, and they also maintained a linear extrapolated $T_{80}$ lifetime of 2600 h under light-dark cycling. This represents the highest performance in all standard stability tests for air-processed PSCs reported to date.

## Results

### Anchoring ability of modifiers on oxide electrodes

We focused on investigating the anchoring capability of modifiers on oxide electrodes. Common modifiers can be primarily categorized into ionic modifiers and molecular modifiers. For ionic modifiers, their binding to the oxide surface does not strictly form ionic bonds but rather primarily involves electrostatic adsorption, which constitutes a physical interaction. In the case of molecular modifiers, their binding to the oxide surface is mainly governed by relatively weak hydrogen-bonding interactions. These interactions are relatively low in energy, making the modifier molecules prone to detachment (Fig. 1a). In contrast, DMF molecules in the liquid phase system contain an amide

group (-CON-), which exhibits strong binding capability and can engage in robust interactions with most modifiers. In addition, DMF is a highly polar solvent, where the modifier-DMF interaction is typically dominated by strong hydrogen bonding, supplemented by ion-dipole interactions, collectively enhancing the dissolution of the modifier in DMF. These two types of bonds represent relatively strong intermolecular forces, resulting in a comparatively higher adsorption energy. Consequently, during device fabrication, modifiers on the oxide surface are highly susceptible to dissolution and removal by perovskite precursor solvents, ultimately leading to the failure of the intended passivation effect[14–17].

To validate these hypotheses, we selected three representative modifiers to evaluate their anchoring effect on oxide electrodes. The structural formulas of these modifiers are presented in Supplementary Fig. 1. Specifically, inorganic salts (potassium chloride, KCl), organic salts (guanidine hydrochloride, $CH_5N_3$·HCl), and ionic liquids (methylammonium acetate, MAAc) were used to modify the $SnO_2$ electrode commonly used in n-i-p type devices. We employed density functional theory (DFT) calculations to determine the binding energies of the aforementioned three modifiers with the electrodes, as well as their binding energies with the typical solvent DMF used in perovskite precursors (Fig. 1b). As shown in Fig. 1c–e, the adsorption energies between all modifiers and DMF were higher than those with the electrodes. We rinsed the modified electrodes with DMF and conducted X-ray photoelectron spectroscopy (XPS) analysis. After rinsing the KCl-modified $SnO_2$ electrode with DMF, the intensity of the K $2p$ peak decreased (Fig. 1f), and the ratio of the K $2p_{3/2}$ peak area to that of the lattice O atoms dropped from 19.0% to 7.7% (Supplementary Fig. 2a and Supplementary Table 1), indicating desorption of KCl from the $SnO_2$ electrode surface. For the $CH_5N_3$·HCl-modified $SnO_2$ electrode, after rinsing with DMF, the intensity of the N $1s$ peak decreased (Fig. 1g), and the ratio of the N $1s$ peak area to that of the lattice O atoms dropped from 22.4% to 15.9% (Supplementary Fig. 2b and Supplementary Table 2), representing desorption of $CH_5N_3$·HCl from the $SnO_2$ electrode surface. In Fig. 1h, after rinsing the MAAc-modified $SnO_2$ electrode with DMF, the intensity of the C $1s$ peak decreased, and the ratio of the C $1s$ peak area to the lattice O atoms decreased from 18.1% to 12.3% (Supplementary Fig. 2c and Supplementary Table 3), indicating desorption of MAAc from the $SnO_2$ electrode surface[17]. These experimental results are consistent with the theoretical calculation results. In addition, contact angle test results also confirmed this conclusion (Supplementary Fig. 3).

### Comprehensive characterization of a fully chemisorbed interface

We strive to minimize the desorption of electrode modifiers without affecting the normal perovskite preparation process. Taking the $SnO_2$ electrode as an example, a fully chemical adsorption method has been proposed. Considering the chelating effect with multiple binding sites, a molecule of DTPA, which is rich in -COOH groups (Supplementary Fig. 4a), was introduced into the $SnO_2$ colloidal dispersion to prepare $SnO_2$ thin films. Subsequently, the surface was modified with an aqueous solution of Zol (Supplementary Fig. 4b), which contains two -PO(OH)$_2$ groups (Supplementary Fig. 5). We speculate that the reaction process is as shown in Fig. 2a. During the heating process at 120 °C, one -PO(OH)$_2$ group in the Zol molecule will undergo an esterification reaction with the -COOH group in DTPA, forming a phosphodiester bond. Furthermore, the other -PO(OH)$_2$ group in the Zol molecule will undergo an esterification reaction with the excess -COOH group in another surrounding DTPA, and numerous phosphodiester bonds will bond the molecules together, thereby forming a robust in-situ cross-linked-like covering modification layer on the $SnO_2$ surface. Supplementary Fig. 6 shows the electrostatic potential of DTPA and Zol. To confirm the specific reactions between DTPA and Zol on the $SnO_2$ surface, liquid-state proton nuclear magnetic resonance

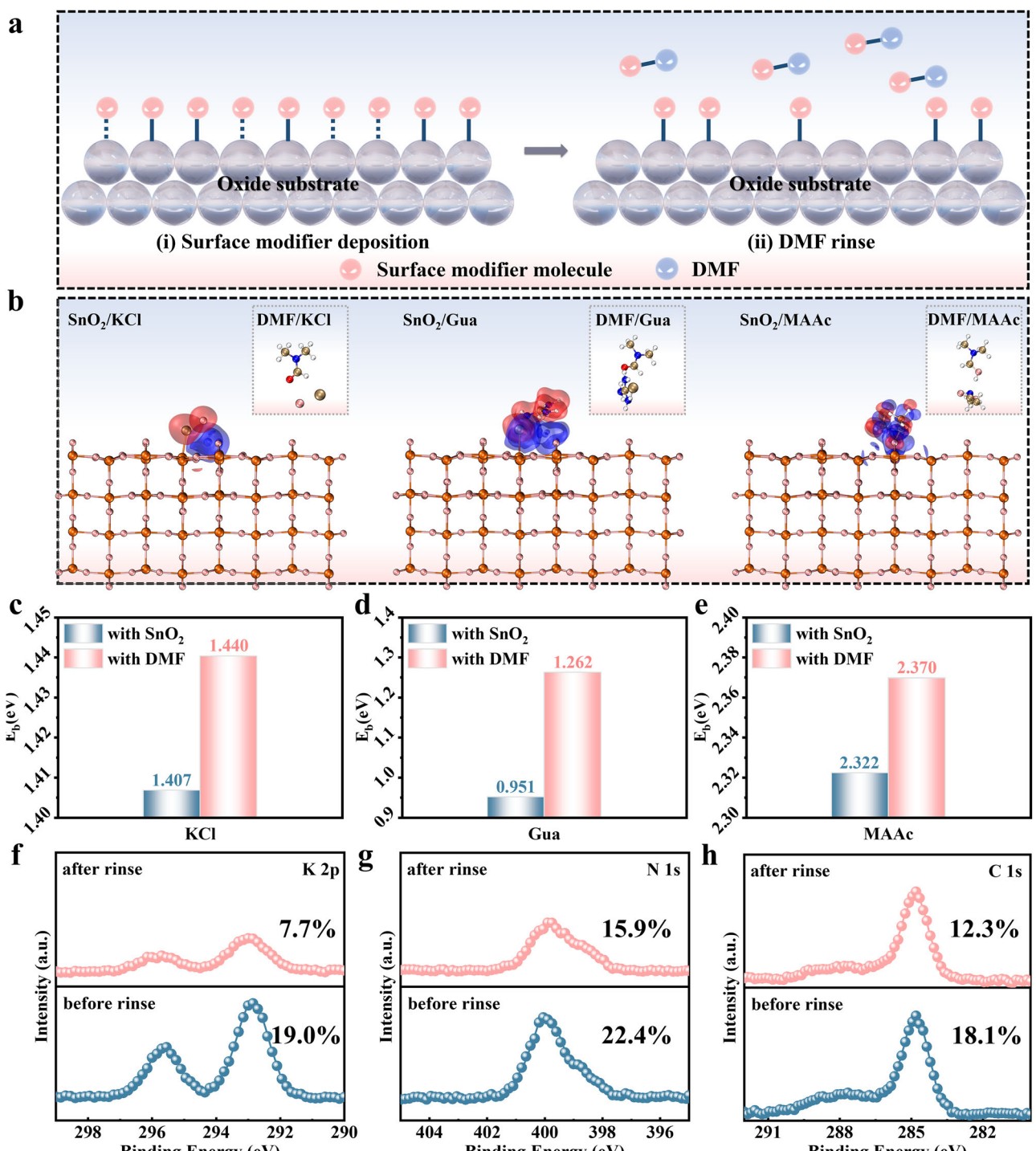

**Fig. 1 | Study on the anchoring ability of modifiers on oxide electrodes.**
**a** Schematic of the adsorption and desorption of modifiers on the oxide electrode surface (before and after washing with DMF). The dashed lines represent weak hydrogen bonds. **b** DFT calculations of the binding energy between the modifier and the oxide electrode. From left to right: SnO₂/KCl, SnO₂/Gua, and SnO₂/MAAc. The inset shows their binding energies with DMF. All relevant visualization and interpretation are assisted by multiwfn[51–53] and VMD[54]. **c–e** Binding energies of the modifier with the oxide and with DMF. **f–h** XPS spectra of each electrode before and after washing with DMF. **f** K 2p core level of SnO₂/KCl. The numbers in the figure represent the ratio of the K 2p peak area to the lattice O peak area. **g** N 1s core level of SnO₂/Gua. The numbers in the figure represent the ratio of the N 1s peak area to the lattice O peak area. **h** C 1s core level of SnO₂/MAAc. The numbers in the figure represent the ratio of the C 1s peak area to the lattice O peak area.

($^1$H NMR) testing was conducted. The samples were dissolved in deuterated dimethyl sulfoxide (DMSO-$d_6$) for verification (Supplementary Fig. 7–9). Supplementary Fig. 9 shows the characteristic $^1$H signal of the phosphodiester bond at δ 3.79 ppm in the product after the reaction between DTPA and Zol at 120 °C. The Fourier-transform infrared spectroscopy (FTIR) in Supplementary Fig. 10 highlights the interaction between -PO(OH)₂ and -COOH groups in SnO₂ electrodes modified with both DTPA and Zol. Specifically, for SnO₂ modified with DTPA, the stretching vibration peak of the C=O bond appears at 1640 cm$^{-1}$, while for SnO₂ modified with Zol, the $P$=O bond exhibits a peak at 1640 cm$^{-1}$. When both modifiers are present, the intensity of the 1640 cm$^{-1}$ peak decreases and exhibits a slight red shift, indicative

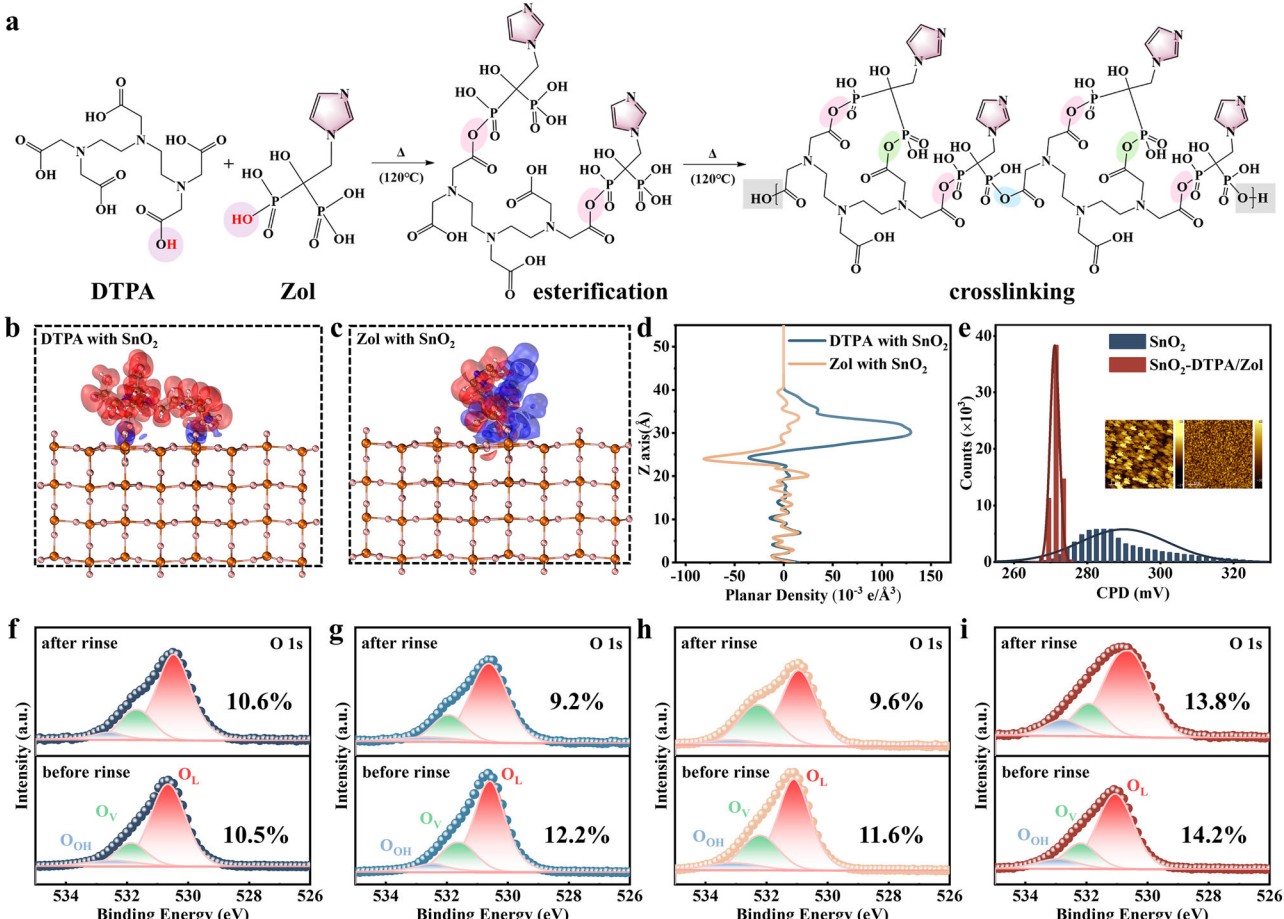

**Fig. 2 | Study on the interfacial properties of chemisorption. a** Schematic of possible chemical reactions of DTPA and Zol. **b** DFT calculations of the binding energy of DTPA with SnO₂. **c** DFT calculations of the binding energy of Zol with SnO₂. **d** Surface integration of the differential charge density obtained from DFT calculations. **e** KPFM measurements showing the CPD distribution of SnO₂ films before and after modification. The inset shows surface potential images of Control (left) and Target (right). **f–i** XPS spectra of the O 1s core level before and after washing with DMF for different SnO₂ electrodes: (**f**) SnO₂, (**g**) SnO₂-DTPA, (**h**) SnO₂/Zol, (**i**) SnO₂-DTPA/Zol. The numbers in the figure represent the ratio of the -OH peak area to the lattice O peak area.

of hydrogen bonding or other interactions between -PO(OH)₂ and -COOH groups. This is consistent with the ¹H NMR results.

DFT was used to calculate the interactions between DTPA and Zol molecules with the SnO₂ electrode (Fig. 2b–d). During the simulation process, both -PO(OH)₂ and -COOH can form coordination bonds with uncoordinated Sn. The binding energy ($E_b$) between DTPA and the top surface of SnO₂ was calculated to be − 0.771 eV, which is lower than the $E_b$ between DTPA and DMF (− 1.140 eV). The $E_b$ between Zol and the top surface of SnO₂ was calculated to be − 0.199 eV, which is lower than the $E_b$ between Zol and DMF (1.171 eV). The differential charge density in Fig. 2d shows significant charge transfer between the two modifier molecules (DTPA and Zol) and SnO₂. The corresponding electron localization function (ELF) images demonstrate electron cloud overlap between Sn and O atoms (Supplementary Fig. 11), further confirming the strong coupling effect between the groups (-COOH in DTPA, -PO(OH)₂ in Zol) and the SnO₂ lattice. These results prove the strong interactions between our selected modifier molecules and the SnO₂ substrate. Changes in elemental binding energies further confirmed the aforementioned mechanism through XPS testing. As shown in Supplementary Fig. 12 and Supplementary Table 4, after modification with DTPA and Zol, the peak of Sn 3d shifted to higher binding energy, and the proportion of the C = O peak decreased, which can be attributed to the chemical bonding between Sn and the negatively charged O in the modifier molecules[24]. In addition, peaks for N 1s and P 2p also appeared after the modification.

In addition, to verify the optimization effect of the modifiers on the SnO₂ surface, Kelvin probe force microscopy (KPFM) and ultraviolet photoelectron spectroscopy (UPS) tests were carried out. Figure 2e shows the change in contact potential difference (CPD) obtained from the KPFM test. The CPD distribution of the films modified with DTPA and Zol became significantly narrower, which implies a reduction in surface defect density, beneficial for effective photocarrier extraction and lower open-circuit voltage ($V_{oc}$) loss[25,26]. UPS demonstrated the impact of the modifiers on the energy level of the SnO₂ surface and calculated the changes in the SnO₂ band structure (Supplementary Fig. 13). Detailed values are listed in Supplementary Table 5. After co-modification with DTPA and Zol, both the valence band maximum (VBM) and conduction band minimum (CBM) of SnO₂ shift upward. Femtosecond transient absorption spectroscopy (fs-TAS) was employed to investigate the charge carrier dynamics between modified SnO₂ and PVK. In both cases, a strong and immediate ground-state bleaching (GSB) peak at ~ 784 nm was observed upon excitation of the host perovskite absorption (Supplementary Fig. 14a, b). Typically, the GSB signal is proportional to carrier density. For the SnO₂/perovskite stack, the variation in GSB peak intensity with delay time directly reflects the quantity of photogenerated carriers in the conduction and valence bands of the perovskite[27]. Compared to pristine SnO₂/PVK, weaker GSB signals at identical delay times were exhibited by the SnO₂-DTPA/Zol/PVK sample (Supplementary Fig. 14c, d), along with a higher electron

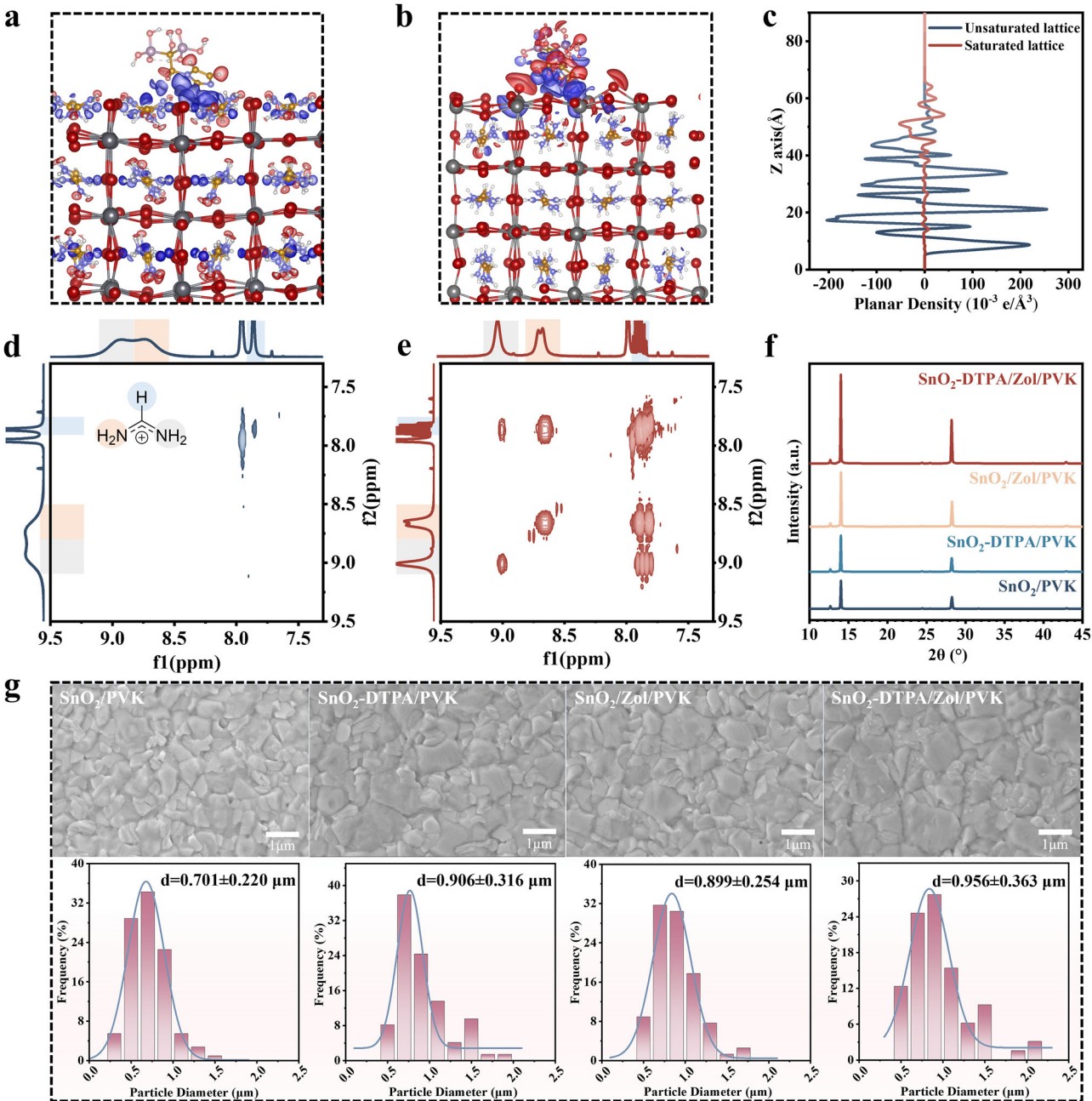

**Fig. 3 | Study on the SnO₂/FAPbI₃ interfacial properties.** DFT calculations of the binding energy between Zol and FAPbI₃ for (**a**), unsaturated lattice and (**b**), saturated lattice. **c** Surface integration of the differential charge density obtained from DFT calculations. **d** 2D ¹H NMR spectrum of FAPbI₃, and (**e**) reaction products of DTPA and Zol, as well as the FAPbI₃ mixture. **f** XRD patterns and (**g**) SEM images (with particle size distribution histogram below) of FAPbI₃ films grown on different SnO₂ electrodes before and after modification.

extraction rate (Supplementary Fig. 14e). It is concluded that faster electron transfer from the perovskite to the adjacent modified SnO₂ contact occurs. This can be attributed to the critical role of interface modification in promoting carrier extraction, and it is consistent with the KPFM. More importantly, we rinsed the electrodes before and after modification with DMF. XPS showed that the electrodes co-modified by DTPA and Zol exhibited good anchoring stability (Fig. 2f–i and Supplementary Table 6). The contact angle tests also yielded consistent results with XPS (Supplementary Fig. 15). Cross-sectional TEM-EDS mapping reveals distinct and homogeneous distribution of P and C elements throughout the SnO₂ layer, demonstrating uniform dispersion of the modifiers within the SnO₂ transport layer(Supplementary Fig. 16).

## Interactions of the perovskite film and the SnO₂-DTPA/Zol interface

We explored the impact of interface modifications at the buried interface. DFT theoretical calculations demonstrated the interaction between the Zol closest to the perovskite film and the bottom terminal of the perovskite (Fig. 3a–c). Figure 3a, b respectively depict the significant differences in charge density between PVK with unsaturated lattice and saturated lattice interacting with Zol molecules. The changes in electron cloud distribution indicate charge gain and loss, with red and blue regions representing electron depletion and accumulation, respectively, due to electron redistribution. In Fig. 3a, the electron density increases on the FA⁺ surface while decreasing on the N atom of the Zol molecule above it, suggesting that Zol bonds with FA⁺ through

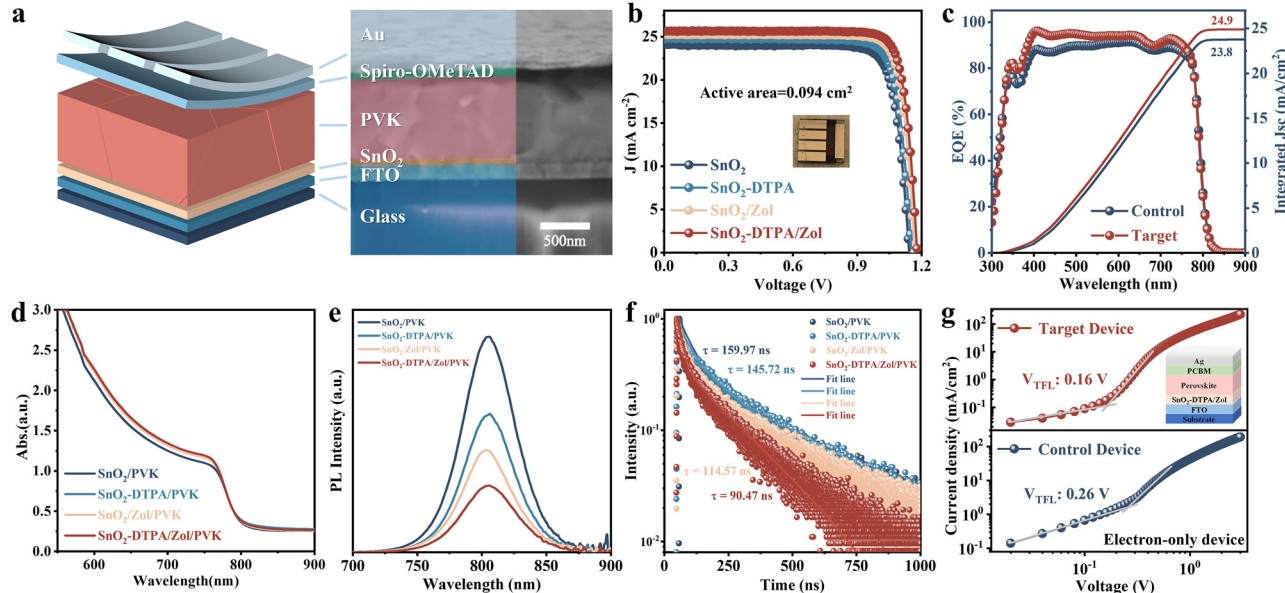

**Fig. 4 | Study on the optoelectronic and photovoltaic performance of PSCs.**
**a** Schematic of the n-i-p PSC device structure and cross-sectional SEM image. **b** J-V curves of PSCs with $SnO_2$, $SnO_2$-DTPA, $SnO_2$/Zol, and $SnO_2$-DTPA/Zol as the ETL (inset shows a photo of the best-performing device). **c** Corresponding EQE spectra and integrated $J_{sc}$. Optical properties of $FAPbI_3$ films grown on different $SnO_2$ electrodes before and after modification: (**d**) UV-vis; (**e**) PL; (**f**) TRPL spectra; and (**g**) SCLC curves.

the N atom, thereby transferring electrons to $FA^+$. Similarly, in Fig. 3b, the electron density increases on the Pb atom surface but decreases on the N atom surface, indicating that Zol can also bind to Pb via the N atom, leading to electron transfer to Pb. This is further reflected in the differential charge density analysis (Fig. 3c), where the PVK surface gains electrons, forming a negative charge center, while Zol loses electrons, creating a positive charge center. Consequently, an efficient carrier transfer is established at the Zol/PVK interface. We found that Zol can both bind with $FA^+$ and form bonds with non-coordinated $Pb^{2+}$. 1H NMR tests (Fig. 3d, e and Supplementary Fig. S17) showed that Zol caused the characteristic signal of $FA^+$ to split, further confirming that Zol can form van der Waals bonds with $FA^+$. The variation of the Zol characteristic signal in Supplementary Fig. 18 confirms the interaction between Zol and uncoordinated $Pb^{2+}$. To assess the improvement of the micromorphology and crystal structure of the perovskite film on the $SnO_2$-DTPA/Zol electrode, we conducted X-ray diffraction (XRD) and scanning electron microscopy (SEM) tests. The XRD spectrum showed that the characteristic peaks of the perovskite film on the $SnO_2$ electrode before and after modification were essentially the same, while the peak intensity of the perovskite film on the $SnO_2$-DTPA/Zol electrode was significantly enhanced (Fig. 3f). This indicates that the atomic arrangement orderliness at the lower interface of the modified perovskite film has been significantly improved. The top-view SEM image confirmed that the grain size of the perovskite film on the $SnO_2$-DTPA/Zol electrode was larger than that of the control film (Fig. 3g). These experimental results prove that the $SnO_2$-DTPA/Zol electrode is beneficial for the crystallization and growth of the $FAPbI_3$ film.

## Photovoltaic performance of PSCs

We fabricated devices with an n-i-p stack of glass/FTO/$SnO_2$/$FAPbI_3$/Spiro-OMeTAD/Au in a full air environment (Fig. 4a). We prepared $FAPbI_3$ using the one-step solution process (Fig. 4b) and the two-step solution process (Supplementary Fig. 19), respectively. Devices with $SnO_2$-DTPA/Zol as the electron transport layer (ETL) exhibited the best photovoltaic performance. At a mask area of 0.094 cm², the PCE reached 25.52% (certified 25.49%, Supplementary Fig. 20), with an open-circuit voltage ($V_{oc}$) of 1.18 V, a short-circuit current density ($J_{sc}$) of 25.59 mA cm⁻², and a fill factor ($FF$) of 84.51% (Supplementary

Table 7). This is one of the highest efficiencies for PSCs prepared in a full-air environment (Supplementary Table 8). These parameters are superior to the control devices (PCE = 22.65%, $V_{oc}$ = 1.14 V, $J_{sc}$ = 24.12 mA cm⁻², $FF$ = 82.50%). At the same time, compared to the control device, the target device has significantly reduced hysteresis (Supplementary Fig. 21). External quantum efficiency (EQE) testing confirmed the measured $J_{sc}$ (Fig. 4c). The EQE of the target devices was consistently higher than that of the control devices across the entire visible light absorption region. By integrating the EQE over the AM 1.5 G standard solar spectrum, the integrated current density $J_{sc}$ for the control and target devices were calculated to be 23.8 mA cm⁻² and 24.9 mA cm⁻², respectively, which closely matched the current density $J_{sc}$ measured under the solar simulator. In addition, the co-modification with Zol and DTPA was also tried in a large-area submodule. The fabricated submodule with an aperture area of 30 × 30 cm² showed a PCE of 16.49% (Supplementary Fig. 22). These results further indicate that DTPA and Zol reduced the defect density at the $SnO_2$/$FAPbI_3$ interface, minimized interface losses, and improved the $V_{oc}$ and PCE of the PSCs.

To further verify the passivation effect of DTPA and Zol on the defects at the $SnO_2$/$FAPbI_3$ interface, we conducted relevant tests on the photoelectronic properties of thin films. The results of ultraviolet-visible absorption (UV-vis) spectra indicated that the absorption intensity of the $SnO_2$-DTPA/Zol perovskite film was slightly higher than that of the initial sample (Fig. 4d). The photoluminescence (PL) spectra described the steady-state optical properties of the perovskite films (Fig. 4e). The fluorescence absorption peak positions of all samples were beyond 800 nm. Compared to the pristine $SnO_2$/PVK sample, the $SnO_2$-DTPA/Zol/PVK sample exhibits significantly reduced photoluminescence peak intensity, demonstrating that the co-modification enhances the electron extraction capability of the $SnO_2$ ETL, thereby facilitating more efficient charge transfer from the perovskite to the ETL[28]. The time-resolved photoluminescence (TRPL) spectra described the carrier dynamics of the perovskite films (Fig. 4f and Supplementary Table 9). The carrier lifetimes ($\tau_1$ = 24.48 ns, $\tau_2$ = 199.22 ns, $\tau_{ave}$ = 90.47 ns) of the perovskite film deposited on $SnO_2$-DTPA/Zol were shorter than those of the control sample ($\tau_1$ = 54.25 ns, $\tau_2$ = 354.69 ns, $\tau_{ave}$ = 159.97 ns). The reduction in carrier lifetime confirmed a significant

enhancement in charge extraction capability at the modified ETL/perovskite interface, with a significant decrease in radiative recombination of carriers[29]. The defect density was assessed in electron-only devices (FTO/ETL/perovskite/PCBM/Ag) using space-charge-limited current (SCLC) (Fig. 4g). The SCLC curve is divided into three regions. In the low-bias left region, the current varies linearly with voltage, which is considered an ohmic contact. Then, as the bias voltage increases, it gradually enters the trap-filling region, where carriers fill traps through continuous injection[30]. The intersection of these two regions is called the trap-filling limit voltage ($V_{TFL}$). The high-bias region of the SCLC curve represents a quadratic relationship between current and voltage[31]. After modifying SnO$_2$ with DTPA and Zol, the defect density ($N_t$) was reduced from $3.68 \times 10^{15}\,cm^{-3}$ to $2.26 \times 10^{15}\,cm^{-3}$, which decreased the defect density caused by interfacial recombination and promoted effective charge extraction and transport. This result is consistent with the PL and TRPL test results.

The Mott-Schottky curves of the PSCs were obtained under dark conditions to explore the mechanisms of $V_{oc}$ and PCE (Supplementary Fig. 23)[32]. The built-in potential of the target devices (0.90 V) is greater than that of the original devices (0.86 V), indicating that the electric field at the modified SnO$_2$/FAPbI$_3$ interface has become stronger. This can promote charge separation, prevent charge accumulation at the interface, and enhance the transport performance of charge carriers. As shown in Supplementary Fig. 24, the formation of dark current is due to the migration of charge carriers caused by defects present in the PSCs[24]. Within the voltage range of $-0.6$ to 0.6 V, the dark current density of the target devices is lower than that of the control devices, further confirming the passivation effect of the modified SnO$_2$/FAPbI$_3$ interface. We also studied the electrochemical performance and charge recombination of the PSCs using electrochemical impedance spectroscopy (EIS) (Supplementary Fig. 25). It was observed that compared to the control devices (2850 Ω), the target devices (4510 Ω) exhibit a higher recombination resistance ($R_{rec}$), indicating a reduction in carrier recombination losses within the PSCs[33].

To investigate the phase stability, we conducted aging tests on FAPbI$_3$ films with different electrodes using an ultraviolet lamp and a hot stage. Supplementary Fig. 26 shows that the initial film almost completely decomposed after aging for 13 h under an ultraviolet lamp at 254 nm ($T_{ambient} = 35\,°C$, RH = 85%), with a significant reduction in the intensity of the characteristic peaks of its black phase in XRD. In contrast, the FAPbI$_3$ film on the SnO$_2$-DTPA/Zol electrode still maintained a clear black phase after aging for 13 h. Similarly, the results after aging at 110 °C ($T_{ambient} = 35\,°C$, RH = 85%, natural light) were consistent with those under the ultraviolet lamp (Supplementary Fig. 27). Furthermore, the microstructural morphology (Supplementary Fig. 28) and optical absorption properties (Supplementary Fig. 29) of all thin films were characterized before and after aging, with the test results being fully consistent with the XRD data. These results are attributed to the fixation of the terminal FA$^+$ cations at the bottom of FAPbI$_3$ by DTPA and Zol, and the repair of interfacial defects, leading to a significant improvement in the phase stability of FAPbI$_3$. Here, we define an environment with constant humidity, temperature, or light conditions as a static environment, while an environment where these conditions change continuously over time is referred to as a dynamic environment. Currently, many aging tests for PSCs in laboratories are conducted under static environmental conditions, yielding promising stability results. However, during outdoor operation, PSCs are subjected to continuously changing humidity, temperature, and light conditions due to seasonal weather variations and the daily cycles of sunrise and sunset. Therefore, it is crucial to investigate both the static and dynamic stability of PSCs. Regarding static stability, the dark storage stability experiment was conducted on devices with an area of 0.094 cm² according to the ISOS-D-1 protocol[34] (Fig. 5a). The unencapsulated devices processed with both DTPA and Zol retained over 97% of their initial PCE after being stored for 2750 h in an air

environment (Relative humidity: 20–40%) at 23 ± 4 °C, while the control devices' PCE dropped to nearly 80% of their initial value under the same conditions. The thermal stability experiment was conducted on devices with an area of 0.28 cm² according to the ISOS-D-2 protocol (Fig. 5c). The unencapsulated target devices retained over 97% of their initial PCE after being treated on a hot plate at 85 °C in a nitrogen environment for 950 h, whereas the control devices' PCE dropped to about 80% of their initial value under the same conditions. For dynamic stability, the light-dark cycling experiment was conducted on devices with an area of 0.28 cm² according to the ISOS-LC-1 protocol (Fig. 5e). Supplementary Fig. 30 shows the light-dark cycling procedure. The unencapsulated target devices retained 93% of their initial PCE after 1008 h, while the control devices' PCE decreased to 67% of their initial PCE. After pre-exposure for 1000 h under damp heat conditions, we further conducted MPP tracking tests on devices with an area of 0.28 cm² according to the ISOS-L-1 protocol to assess the operational stability under harsh conditions (Fig. 5g). The damp heat treatment induced pre-aging of the FAPbI$_3$ film, and after the start of the MPP tracking experiment, illumination accelerated the phase transformation and decomposition of FAPbI$_3$ within the devices[35]. Consequently, the PCE of the control devices rapidly decreased to below 60% of their initial performance after 500 h. However, the target devices were still able to retain over 80% of their PCE. Thermal cycling experiments were performed on devices with an active area of 0.04 cm² (Supplementary Fig. 31). Each cycle consisted of: (1) aging the devices in a vacuum drying oven at 65 °C for 24 h, followed by (2) storage in a N$_2$ glovebox under dark conditions at room temperature for 24 h. After 960 h, the unencapsulated target devices retained over 80% of their initial PCE, while the control devices degraded to below 40% of their initial PCE. Real outdoor stability testing was also performed on 0.04 cm² devices (Supplementary Fig. 32). The encapsulated devices were aged under actual outdoor conditions. After 500 h, the target devices maintained approximately 80% of their initial PCE, whereas the control devices degraded to less than 20% of their initial PCE. It is worth mentioning that we performed linear extrapolation[19,36] on the data from dark storage, thermal stability, and light-dark cycling tests, and derived the theoretical T$_{80}$ values of approximately 27,000 h, 19,000 h, and 2600 h, respectively (Fig. 5b, d, and f). As far as we know, these results position them among the most statically and dynamically stable PSCs[37] (Supplementary Tables S10–12).

## Discussion

In summary, we have demonstrated that the co-decoration of SnO$_2$ electrodes with DTPA and Zol can improve the quality of the SnO$_2$/FAPbI$_3$ interface while ensuring that the modifiers do not detach or fail due to weak chemical bonding. We have elucidated that the mechanism of action is related to the in situ reaction of the modifiers on the SnO$_2$ surface. By pre-embedding DTPA molecules within the SnO$_2$ film and then introducing Zol to the film surface, the two modifiers undergo an in situ cross-linking-like reaction at high temperatures of 120 °C, thereby forming a robust chemical coverage area between the oxide electrode and FAPbI$_3$. At the same time, the part of Zol closest to the perovskite film can bond with FA$^+$ cations, effectively preventing the escape of perovskite terminal cations and reducing defect-induced interfacial recombination losses. Therefore, this strategy can achieve a PCE of 25.52% under standard conditions, one of the highest PCEs for devices fully fabricated in an ambient atmosphere. These resulting unencapsulated cells also exhibit exceptional stability, successfully passing multiple standard stability tests, including dark storage, thermal stability, light-dark cycling, and MPP tests. In addition, we have predicted, using linear extrapolation, that the T$_{80}$ storage lifetime is ≈ 27,000 h, the storage life at 85 °C is ≈ 19,000 h, and the light-dark cycle lifetime is ≈ 2600 h. To our knowledge, this represents the highest combined performance in all standard stability tests for PSCs fabricated by the air method reported to date.

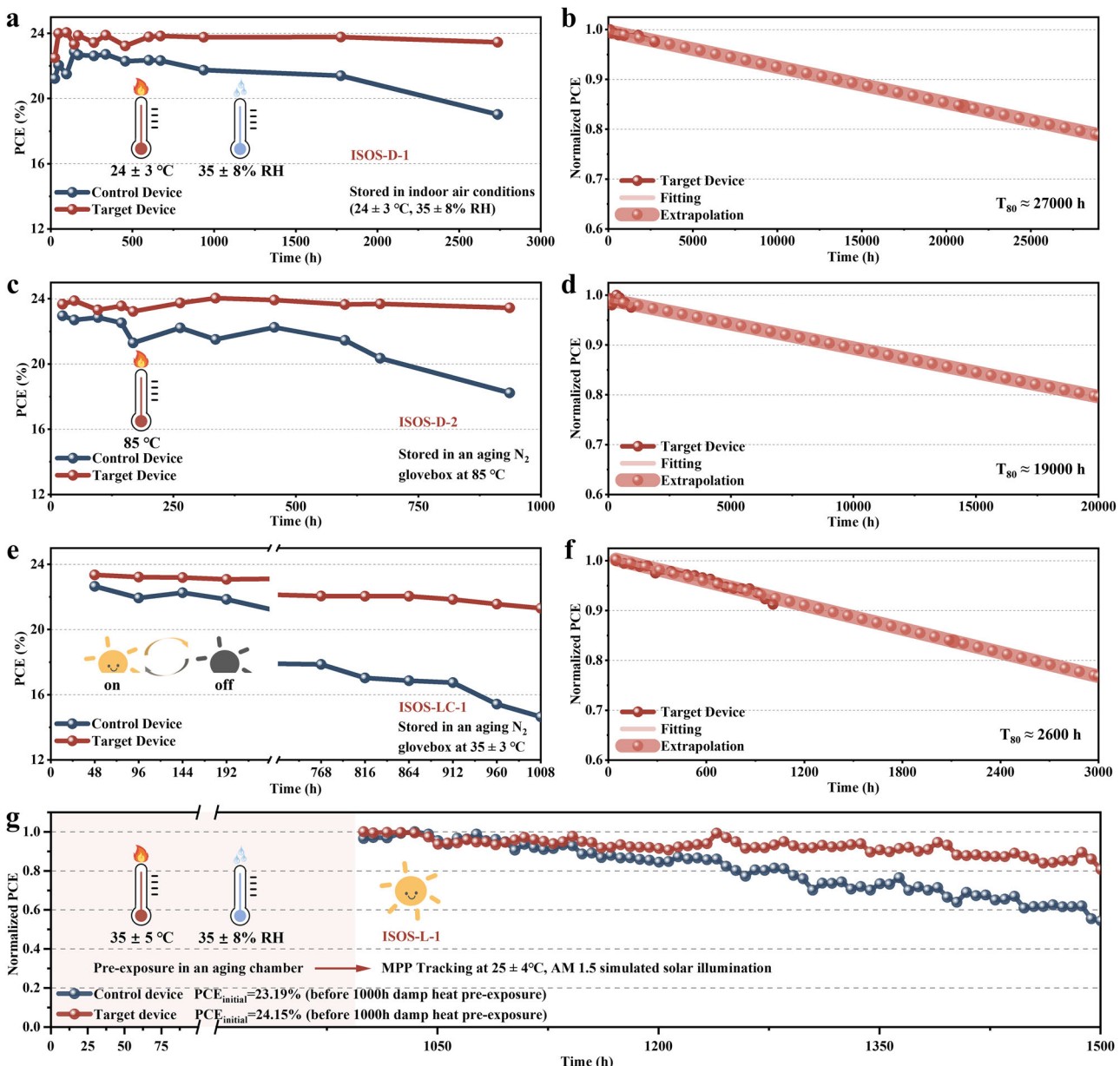

**Fig. 5 | Study on the Stability of PSCs. a** Stability study of PSCs. Evolution of the efficiency of PSCs under dark storage conditions in an indoor air environment (ISOS-D-1). **c** Evolution of the efficiency of PSCs under heating conditions at 85 °C (ISOS-D-2). **e** Efficiency evolution of PSCs under light-dark cycling tests (12 h on/12 h off) (ISOS-LC-1). **b**, **d**, and **f** Corresponding linear extrapolation of lifetime results. **g** MPP tracking of PSCs after 1000 h of damp heat pre-treatment.

## Methods

### Materials

All materials have not been further purified since purchase. FTO/glass and Methylammonium iodide (MAI, > 99.99%) were obtained from Advanced Election Technology. SnO₂ colloid precursor (tin(IV) oxide, Lead (II) iodide (PbI₂, 99.999%), Phenethylammonium iodide (PEAI, 99.5%), Spiro-OMeTAD (99.5%), PTAA (MOS) were purchased from Xi'an Polymer Light Technology Corp. Formamidinium iodide (FAI, > 99.99%) and Methylammonium chloride (MACl, > 99.99%) were purchased from Greatcell Solar Materials Pty Ltd. N, N-dimethylformamide (DMF), dimethyl sulfoxide (DMSO), isopropanol (IPA) and chlorobenzene (CB), 4-tert-butylpyridine (tBP), acetonitrile (Ace), bis(trifluoromethylsulfunyl)-imide lithium salt (Li-TFSI; 99.95%), were obtained from Sigma-Aldrich. FK209 Co(III) TFSI were purchased from Luminescence Technology Corp. Diethylene-triaminepentaacetic acid (DTPA, 99%), Zolephonic acid (Zol, 99%) were purchased from Aladdin.

### Device fabrication

SnO₂ Substrate Preparation: The FTO glass was successively cleaned in detergent, deionized water, and ethanol for 30 min by an ultrasonic cleaner. The clean FTO was treated with an ultraviolet ozone machine for 30 min. The SnO₂ solution was mixed with water at a volume ratio of 1:3 and spin-coated on the FTO at 3000 rpm for 30 seconds, followed by annealing at 150 °C for 30 min in air. For the target devices, 0.1 mM DTPA was pre-added to the SnO₂ solution. After the annealing of SnO₂ film, a Zol solution (2 mM Zol in ultrapure water) was spin-coated onto the SnO₂ film at 3000 rpm for 30 seconds, followed by annealing at 120 °C for 5 minutes.

PSCs Fabrication: The whole process of the PSCs fabrication was carried out in an ambient air environment (room temperature and 20–50% relative humidity). Perovskite films were fabricated by either one-step spin-coating or two-step spin-coating. For surface passivation, 4 mg/mL PEAI in isopropanol was spin-coated on the perovskite film at 5000 rpm for 30 s without further treatment. The hole transfer

material (HTM) solution was prepared by mixing 90 mg Spiro-OMe-TAD, 35 μl bis(trifluoromethane) sulfonimide lithium salt (Li-TFSI) solution (260 mg Li-TFSI in 1 ml acetonitrile), 30 μl 4-tertbutylpyridine (tBP), and 30 μl of FK209 Co(III) TFSI salt (375 mg/mL in acetonitrile). Then, the HTM solution was deposited on the top at 3000 rpm for 30 s in air without annealing. Finally, 80 nm Au was deposited to form an electrode by thermal evaporation under $5 \times 10^{-4}$ Pa. Before the J-V test, a commercial anti-reflection layer (Shengyu Tech.) is carefully stuck on the glass side for better light transmittance.

(1) The perovskite films, which were fabricated by the two-step solution process:

Firstly, 692 mg $PbI_2$ was dissolved in 1 ml DMF: DMSO (v: v = 9:1), spin-coated on the $SnO_2$ electrode at 1500 rpm for 30 s, and annealed at 70 °C for 1 min. After cooling down to room temperature, the organic solution (92.5 mg FAI, 6.5 mg MAI, and 9.5 mg MACl in 1 mL IPA, stirred for more than 2 h) was spin-coated onto the $PbI_2$ film at 2500 rpm for 3 s, 1350 rpm for 15 s and 1700 rpm for 12 s. The film was annealed at 150 °C for 15 min. All spin-coating and annealing processes were performed under ambient air conditions (room temperature and 20–50% R.H.).

(2) The perovskite films, which were fabricated by the one-step solution process[38]:

The perovskite precursor solution was prepared by dissolving 810 mg of black-phase $FAPbI_3$, 35 mol% of MACl, and 1 mol% of acetylcholine chloride in DMF and DMSO (8:1 v/v) at 60 °C. The perovskite layers were then spin-coated at 1000 rpm for 10 s and 5000 rpm for 15 s, and 1 mL of ethyl ether was dripped onto the electrode during spin-coating. The perovskite layers were annealed at 120 °C for 40 min, and the electrode was cooled. All spin-coating and annealing processes were performed under ambient air conditions (room temperature and 20–50% R.H.).

(3) The stability analysis devices:

Under heating conditions, PEAI could be gradually converted into the $PEA_2PbI_4$ phase, leading to reduced device operational lifetime[39]. Consequently, we removed the PEAI passivation layer for all thermal-related stability tests. In addition, given the intrinsic instability of Spiro-OMeTAD as an HTL[40–42], we adopted a Spiro-OMeTAD/PTAA hybrid HTL for most stability evaluations, while employing pure PTAA HTL for a limited number of tests to ensure enhanced stability.

Specifically, for thermal stability testing, the perovskite films received no surface modification with PEAI or any other agent. The Spiro-OMeTAD/PTAA mixed solution (90/15 mg/mL in chlorobenzene) was supplemented with 35 μL Li-TFSI solution (260 mg/mL in acetonitrile), 30 μL tBP, and 30 μL FK209 Co(III) TFSI salt (375 mg/mL in acetonitrile), followed by spin-coating deposition at 5000 rpm for 30 s.

For storage, MPPT, and light-dark cycling tests, the perovskite films were modified with OAI (5 mg in 1 mL isopropanol) through spin-coating at 4000 rpm for 30 s and subsequently annealed at 150 °C for 10 min to form a 2D perovskite surface layer. The Spiro-OMeTAD/PTAA mixed solution (90/7 mg/mL in chlorobenzene) was supplemented with 35 μL Li-TFSI solution (260 mg/mL in acetonitrile), 30 μL tBP, and 30 μL FK209 Co(III) TFSI salt (375 mg/mL in acetonitrile), and deposited via spin-coating at 5000 rpm for 30 s.

Regarding thermal cycling and outdoor storage stability tests, the perovskite films remained unmodified with PEAI or other surface treatments. The PTAA solution (20 mg/mL in chlorobenzene) was mixed with 10 μL tBP and 2.6 μL Li-TFSI solution (260 mg/mL in acetonitrile), then spin-coated at 5000 rpm for 30 s.

## Calculation

The ground-state geometries of the investigated molecules were optimized using the B3LYP functional with the 6-311 G (d, p) basis set[43–48]. The absence of imaginary frequencies in the optimized structures confirms that all geometries correspond to energy minima.

Density functional theory (DFT) calculations were performed using the Gaussian 09 program[49].

The perovskite ($FAPbI_3$)/zol interface system was optimized using the generalized gradient approximation (GGA) with the Perdew-Burke-Ernzerhof (PBE) exchange-correlation functional in CP2K[50]. The corresponding input files are constructed with the assistance of multiwfn. Calculations identify the $FAPbI_3$ (001) plane as the most stable surface. To reduce computational costs, a $4 \times 4 \times 3$ $FAPbI_3$ perovskite supercell was constructed. The electron wave functions were expanded using a plane-wave basis set with a kinetic energy cutoff of 400 eV. For K-point sampling, a $1 \times 1 \times 1$ Monkhorst-Pack grid was applied in the irreducible Brillouin zone. A 20 Å vacuum region was added to the outer $PbI_2$ surface to avoid spurious interactions. A similar treatment was performed on the exposed lattice to study its interaction with FA cations. To standardize the initial configuration, all molecules were placed equidistantly on the perovskite surface. Self-consistent iterative convergence was achieved at a threshold of $1.0 \times 10^{-4}$ eV/atom, and atomic positions were relaxed until the maximum force on each atom was below 0.05 eV/Å.

Similar methods were used to optimize and calculate the electron layer/modifier interface systems. According to the literature, tin oxide ($SnO_2$) was modeled using the [110] plane, while nickel oxide (NiO) was modeled using the [100] plane. To reflect practical conditions, oxygen vacancies were introduced into the electron layers as appropriate[11,24].

The binding energy of DMF/modifier interactions was calculated based on electron layer/modifier simulations. The adsorption energy ($E_{ads}$) was determined using the formula:

$$E_{ads} = E_{Total} - E_{base} - E_{app} \tag{1}$$

Here, $E_{Total}$ is the total energy of the system, $E_{base}$ is the energy of the base material, and $E_{app}$ is the energy of the adsorbed molecule or modifier.

## Characterization

The current density–voltage (J–V) curves were measured using an AM 1.5 G solar simulator equipped with a Xenon lamp (USHIO) and a Keithley 2450 source meter. The light intensity was calibrated to be 100 mW/cm² using a NIST-certified monocrystalline Si solar cell. For all measurements, a non-reflective metal mask with an aperture area of 0.094 cm² was used to cover the active area of the device to avoid light scattering through the sides. The steady-state photoluminescence (PL) and time-resolved photoluminescence (TRPL) were performed via 450 nm laser in FLS 1000, Edinburgh Instruments. The surface chemical environment of perovskite films was obtained by X-ray photoelectron spectrometer with binding energy referenced to C 1 s peak at 284.8 eV (XPS, Thermo Scientific ESCALAB Xi +). The Nuclear magnetic resonance ($^1H$-NMR) spectra were measured with a 400-MHz spectrometer (BRUKER AVANCE NEO 400 M). Ultraviolet-visible (UV-vis) spectra were measured with a Varian Cary 5. The SEM images of the perovskite films were measured by a field-emission scanning electron microscope (FE-SEM, Zeiss Sigma 300). The KPFM images of the perovskite films were taken by Horiba JY Labram EVO. The Mott-Schottky curves and EIS data were measured by an electrochemical workstation (AMETEK PARSTAT 3000-DX). Crystal structure information was gathered using a powder X-ray diffractometer (PXRD, Bruker D8 Advance) equipped with a Cu Kα radiation source. Additional information on structures and chemical bonding was measured using Fourier-transformed infrared spectroscopy (FTIR, Thermo Scientific Nicolet iS50).

## Reporting summary

Further information on research design is available in the Nature Portfolio Reporting Summary linked to this article.

## Data availability

The data that supports the findings of the study are included in the main text and supplementary information files or upon request from the corresponding authors. Source data are provided in this paper.

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

## Acknowledgements

This work was supported by financial support from the National Natural Science Foundation of China (52203359). This work was also supported by the National and Jiangsu Province NSF (T2293691, BK20212008) of China, National Key Research and Development Program of China (2019YFA0705400), the Research Fund of National Key Laboratory of Mechanics and Control for Aerospace Structures (MCMS-I-0422K01), the Fundamental Research Funds for the Central Universities (NC2023001, NJ2023002, NJ2022002) and the Fund of Prospective Layout of Scientific Research for NUAA (Nanjing University of Aeronautics and Astronautics). We acknowledge the characterization support of the Center for Microscopy and Analysis at Nanjing University of Aeronautics and Astronautics. The research work is supported by the supporting funds for talents of Nanjing University of Aeronautics and Astronautics. J.P. and S.I.S. acknowledge support from the Basic Science Research Leader Program (Grant NRF-2018R1A3B1052820) through the National Research Foundation of Korea (NRF) funded by the Ministry of Science, ICT & Future Planning (MSIP).

## Author contributions

Riming Nie conceived and directed the project. Luyao Li, Riming Nie, Sang Il Seok, Lixiong Yin and Wanlin Guo reviewed the experiment, analyzed the data, and wrote the manuscript, and all authors discussed the results and commented on the manuscript. Luyao Li fabricated films/ devices and conducted XRD/UV-vis/SEM/XPS/UPS/PL/TRPL/FTIR/Photovoltaic performance characterization/stability measurements. Cheng Wang performed density functional theory calculations. Jaewang Park and Weicun Chu helped fabricate and characterize the devices. Weicun Chu conducted KPFM characterizations. Weicun Chu and Yiming Dai helped carry out the EQE/SCLC/Dark J-V curve and analyze data. Qiankai Ba and Kaifeng Wang helped in fabricating the large-area submodule and tested the efficiency. Cheng Wang and Jiaxing Gao carried out the NMR measurement. Yiming Dai and Jiaxing Gao helped collect stability test data. Zeliang Wei helped collect and analyze contact angle test and J-V curve data. Xiaoming Zhao and Xuchen Nie revised the manuscript.

## Competing interests

The authors declare no competing interests.
