## [Transparent Peer Review file · Nature Communications]

Fully chemical interface engineering for statically and dynamically stable perovskite solar cells

Corresponding Author: Professor Riming Nie

Version 0:

Reviewer comments:

Reviewer #1

(Remarks to the Author)

In this work, Li et al. developed a chemical crosslinking strategy capable of penetrating metal oxide transport layers, enabling strong anchoring of modifiers on the substrate. This approach enhances the quality of the heterointerface between the perovskite and transport layers. By introducing DTPA and Zol molecules into the interior and surface of the transport layer, respectively, the authors demonstrated the existence of chemical crosslinking and the improvement in anchoring capability. The fabricated perovskite solar cells (PSCs) achieved a PCE of over 25%, along with excellent environmental stability under various conditions. While a wide range of experimental techniques was employed to clarify the observations, there remains a gap in the comprehensive explanation and validation of the presented results. Additionally, the following questions and corrections must be addressed in the revised version before publication:

1. The authors calculated the adsorption energies of four modifiers with the oxide as well as with DFM using DFT. Why is the adsorption energy with DFM significantly higher? What is the underlying mechanism behind this phenomenon?
2. In Figures 1C-J, the authors used XPS to demonstrate the desorption of modifiers. Specifically, in Figures 1I and J, they utilized the ratio of adsorbed -OH groups to lattice O atoms in the O 1s peak to reflect the desorption of modifiers. However, based on the provided spectra, the changes appear to be insignificant. Additionally, according to Tables S1-S3, the trends in lattice oxygen values before and after washing are inconsistent. For example, the lattice oxygen peak area decreases after washing for SnO₂/KCl and SnO₂/Gua samples, while it increases for SnO₂/MAAc and NiO/SAM samples.
3. The authors should provide more details and discussion regarding the contact angle test (Figures S2 and S12). The test process is not clearly described, and the results obtained are not sufficiently explained.
4. In Figure S10, why did the N 1s and P 2p peaks disappear for the SnO₂-DTPA sample?
5. Figure 2D presents the differential charge density, but the authors did not discuss this in the main text. It is unclear what specific result or insight the authors aim to convey through this figure.
6. In Figures 2F-I, the -OH peak area for the pure SnO₂ sample after washing reaches ~10.5%, which is higher than that of the SnO₂-DTPA and SnO₂/Zol samples. This appears to be unusual and warrants further explanation.
7. Similarly, the DFT results presented in Figures 3B-E and the NMR test results in Figures S13-S14 require more detailed descriptions and thorough discussion to fully support the findings. If the authors are constrained by space limitations, they may consider adding these details to the Supporting Information (SI).
8. The authors claim that the XRD peak intensity increases after modification, but they did not provide a clear conclusion regarding what this intensity enhancement signifies. It is important to note that peak intensity can be influenced by multiple factors. Additionally, it remains unclear why the modification is beneficial for the crystallization and growth of the FAPbI₃ film. Further explanation is needed to clarify these points.
9. Although the authors confirmed the occurrence of chemical crosslinking between DTPA and Tol molecules through extensive spectroscopic analysis, there is a lack of visual evidence (e.g., TEM, DOI: 10.1002/adma.202300403; DOI: 10.1038/s41467-018-06204-2) to demonstrate whether this linking penetrates the entire oxide transport layer.
10. In Figure 4E, why did the PL peak of the SnO₂-DTPA-perovskite sample exhibit a red shift compared to the other samples?
11. In Figure 4F, the TRPL curves normalized? Why is the fitted parameter A1 greater than 1?
12. The authors used XRD to test stability of films (Figures S21 and S22), why did the PbI₂ peaks for some samples exhibit a shift?

13. Why did the stability tests in Figures 3C, E, and G not display the initial values? The initial efficiency of the devices should be clearly indicated.

14. The authors used Spiro-OMeTAD as the hole transport layer (HTL) in PSCs for both PCE and stability tests. It should be noted that Li-doped Spiro-OMeTAD is inherently unstable, particularly under humidity, light, and thermal conditions. Some of the important references related to this point are recommended for citation and discussion (DOI: 10.1126/science.abo27; DOI: 10.1038/s41467-024-52199-4; DOI: 10.1002/anie.202316183).

15. The authors emphasize the importance of dynamic stability testing, but the manuscript only provides light cycling stability data. Additional dynamic stability tests, such as thermal cycling and even real outdoor stability, should be included to comprehensively evaluate the performance.

16. In the Experimental Section, more details are necessary regarding certain preparation and testing procedures. For example, the authors claim that the perovskite was prepared in air, but specific details are not provided. Additionally, more information should be included about the stability testing protocols to ensure reproducibility and clarity.

Reviewer #2

(Remarks to the Author)

In this manuscript, Li et. al. reports a fully chemical modification strategy in which the interfacial modifiers undergo an in-situ crosslinking-like reaction, forming a localized, chemically bonded layer that seamlessly extends from the bulk of the underlying transport layer to the interface. Perovskite solar cells (PSCs) fabricated with this fully chemical modification strategy achieved a power conversion efficiency (PCE) of 25.52% under standard conditions, representing a high PCE for devices fully fabricated in an ambient atmosphere. In addition, these cells also show excellent stability. This work is interesting and well structured. However, some issues should be addressed before further consideration. Here are the specific comments:

1. The author emphasizes that the use of DTPA and Zol as modifiers can significantly enhance the extraction and migration of interfacial charges. However, the manuscript lacks direct experimental evidence for the kinetic recombination process of photogenerated carriers at the bottom of the perovskite. It is recommended to supplement with femtosecond time-resolved transient absorption spectroscopy (fs-TA) to provide more direct real-time evidence.
2. This study confirms that the in-situ crosslinking-like reaction enabled by the two modifiers, DTPA and Zol, can significantly prevent the desorption of the modifiers. However, the mechanistic explanation for this phenomenon is not yet entirely clear. Please clarify it.
3. In Fig. 3F, the characteristic peaks of PbI₂ in the XRD pattern are high, which seems inconsistent with high efficiency PSCs devices. Please confirm it.
4. In Fig. S21 and S22, the XRD pattern have been provided to analyze the phase stability of FAPbI₃ films. However, more evidence such as the SEM image of different samples should also be given to further prove the improvement of stability of perovskite films.
5. The authors mention the concepts of “statically and dynamically stable” in the title and on lines 282-284 of the manuscript. It is recommended to describe this concept in the Introduction section so that readers can better understand it.
6. As far as we know, PEAI modification alone seems insufficient to achieve such remarkable stability. Did the authors use any other modifiers during the MPPT and thermal stability tests? If so, please supplement detailed descriptions so that readers can reproduce the experiment.

Reviewer #3

(Remarks to the Author)

See attachment

Reviewer #4

(Remarks to the Author)

Reviewer #5

(Remarks to the Author)

Version 1:

Reviewer comments:

Reviewer #1

(Remarks to the Author)

The authors have addressed my concerns, I recommend its publication as it is.

Reviewer #2

(Remarks to the Author)

The authors have adequately addressed all my concerns. I recommend acceptance in its current form.

Reviewer #3

(Remarks to the Author)

The authors have sufficiently addressed my comments. Therefore, I recommend the publication of the manuscript in its current form.

Reviewer #4

(Remarks to the Author)

Reviewer #5

(Remarks to the Author)

Response to reviewers

We would like to thank the reviewers for their valuable comments. These have helped us greatly in improving the quality of our manuscript. We have carefully revised the manuscript to enhance its clarity and facilitate the understanding of the readers. Our point-to-point responses are presented in the following. We hope that the revision would satisfactorily address the comments and concerns of the editors and reviewers.

Reviewer #1

In this work, Li et al. developed a chemical crosslinking strategy capable of penetrating metal oxide transport layers, enabling strong anchoring of modifiers on the substrate. This approach enhances the quality of the heterointerface between the perovskite and transport layers. By introducing DTPA and Zol molecules into the interior and surface of the transport layer, respectively, the authors demonstrated the existence of chemical crosslinking and the improvement in anchoring capability. The fabricated perovskite solar cells (PSCs) achieved a PCE of over 25%, along with excellent environmental stability under various conditions. While a wide range of experimental techniques was employed to clarify the observations, there remains a gap in the comprehensive explanation and validation of the presented results. Additionally, the following questions and corrections must be addressed in the revised version before publication:

Our Response: We thank to the reviewer very much for overall positive comments.

1# The authors calculated the adsorption energies of four modifiers with the oxide as well as with DMF using DFT. Why is the adsorption energy with DMF significantly higher? What is the underlying mechanism behind this phenomenon?

Our Response: Thank you very much for this comment. We have supplemented the relevant content in the main text (Line 96-109, Page 5). DFT calculations reveal that the adsorption energy of modifiers on DMF solvent molecules is higher than that on the oxide substrate. We speculate that this may be attributed to the following factors:

1) For ionic modifiers, their binding to the oxide surface does not strictly form ionic bonds but rather primarily involves electrostatic adsorption, which constitutes a physical interaction. For molecular modifiers, their binding to the oxide surface is mainly governed by weak hydrogen bonding interactions. These interactions are relatively low in energy, making the modifier molecules prone to detachment.

2) In the liquid phase system, DMF molecules contain an amide group (-CON-), which exhibits strong binding capability and can engage in robust interactions with most modifiers. Additionally, DMF is a highly polar solvent, where the modifier-DMF interaction is typically dominated by strong hydrogen bonding, supplemented by ion-dipole interactions, collectively promoting the dissolution of the modifier in DMF. These two types of bonds represent relatively strong intermolecular forces, resulting in a comparatively higher adsorption energy.

(References: DOI: 10.1002/ceat.201500745; DOI: 10.1002/aic.14838; DOI: 10.1038/s41586-021-03964-8; DOI: 10.1126/science.adj9602)

2# In Figures 1C-J, the authors used XPS to demonstrate the desorption of modifiers. Specifically, in Figures 1I and J, they utilized the ratio of adsorbed -OH groups to lattice O atoms in the O 1s peak to reflect the desorption of modifiers. However, based on the provided spectra, the changes appear to be insignificant. Additionally, according to Tables S1-S3, the trends in lattice oxygen values before and after washing are inconsistent. For example, the lattice oxygen peak area decreases after washing for SnO₂/KCl and SnO₂/Gua samples, while it increases for SnO₂/MAAc and NiO/SAM samples.

Our Response: Thank you very much for this comment. We have systematically re-fitted and recalibrated the XPS spectra data for all samples. Specifically, the desorption behavior of modifiers was quantitatively evaluated using the normalized ratio between characteristic element peak areas (of modifiers) and the lattice oxygen peak area. As illustrated in Figure 1 F-H and Figure R1 A-C, the K 2p, N 1s, and C 1s peaks were employed to represent the contents of KCl, Gua, and MAAc, respectively. After intensive DMF washing, all samples exhibited significant reductions in these normalized ratios, unequivocally demonstrating substantial desorption of all three modifiers.

For SnO₂ thin films, the atomic ratios of lattice oxygen (O_L), oxygen vacancies (O_V), and hydroxyl groups (O_{OH}) derived from O 1s peak deconvolution serve as direct indicators of surface quality. To eliminate inter-sample measurement variations, all O 1s fitting results were consistently expressed as atomic ratios. The complete O 1s spectra are provided in Figure S2 A-C and Figure R1 D-F, with detailed quantitative data compiled in Table S1-S3 and Table R1-3. Post-DMF rinse, all substrates showed decreased O_L proportions accompanied by increased O_V and O_{OH} components. This observation confirms the loss of passivation efficacy of these modifiers on SnO₂ surface oxygen vacancy defects, providing conclusive evidence for modifier desorption.

In addition, since the **Reviewer #3** suggested that it would be better to focus solely on SnO₂ in this section, as the inclusion of NiO_x/SAM may cause confusion, we have removed all content related to NiO_x/SAM in this section.

Figure R1. XPS spectra of each electrode before and after washing with DMF.

(A) K 2p core level of SnO₂/KCl. The numbers in the figure represent the ratio of the K 2p peak area to the lattice O peak area. (B) N 1s core level of SnO₂/Gua. The numbers in the figure represent the ratio of the N 1s peak area to the lattice O peak area. (C) C 1s core level of SnO₂/MAAc. The numbers in the figure represent the ratio of the C 1s peak area to the lattice O peak area. O1s core level of (D) SnO₂/KCl, (E) SnO₂/Gua, (F) SnO₂/MAAc.

Table R1. The calculation data for relative content of KCl on SnO₂ substrate surface.

Sample	DMF rinse	K 2p peak area	O 1s peak atomic ratio after peak fitting (%)			Ratio
			Lattice oxygen	Oxygen vacancies	-OH	
SnO ₂ /KCl	Before	12195.71	75.08	19.04	5.88	18.98
SnO ₂ /KCl	After	4958.62	73.13	20.64	6.22	7.70

Table R2. The calculation data for relative content of CH₅N₃·HCl on SnO₂ substrate surface.

Sample	DMF rinse	N 1s peak area	O 1s peak atomic ratio after peak fitting (%)			Ratio
			Lattice oxygen	Oxygen vacancies	-OH	
SnO ₂ /Gua	Before	12807.32	71.71	21.18	3.75	22.41
SnO ₂ /Gua	After	8540.78	70.34	22.06	7.61	15.9

Table R3. The calculation data for relative content of MAAc on SnO₂ substrate surface.

Sample	DMF rinse	C 1s peak area	O 1s peak atomic ratio after peak fitting (%)			Ratio
			Lattice oxygen	Oxygen vacancies	-OH	
SnO ₂ /MAAc	Before	7388.39	73.49	22.97	3.54	18.07
SnO ₂ /MAAc	After	6787.92	71.81	23.30	4.89	12.36

3# The authors should provide more details and discussion regarding the contact angle test (Figures S2 and S12). The test process is not clearly described, and the results obtained are not sufficiently explained.

Our Response: Thank you very much for this comment. As you pointed out, the discussion and analysis of contact angle measurements in the original manuscript were indeed insufficient. We have now supplemented detailed descriptions of the contact angle testing procedures and analysis in the Supporting Information (Figure S3, Figure S15). The specific additions are reproduced below:

The contact angle measurements were performed using ultrapure water droplets as the testing liquid. Figure S3 A-C present contact angle measurements for different SnO₂ substrates: (A) SnO₂/KCl, (B) SnO₂/Gua, and (C) SnO₂/MAAc. All substrates initially exhibited excellent wettability. These substrates were then subjected to extensive DMF washing using the following protocol: 1000 μL of DMF was deposited onto each SnO₂ substrate, allowed to stand for approximately 15 seconds for complete wetting, followed by spin-coating at 3000 rpm for 30 seconds, and finally heated at 150°C for 5 minutes to remove residual solvent. Figures S3 D-F show the corresponding contact angles after intensive DMF washing for (D) SnO₂/KCl, (E) SnO₂/Gua, and (F) SnO₂/MAAc. Notably, all substrates demonstrated significantly increased contact angles post-washing. This phenomenon can be primarily attributed to: The hydrophilic/polar nature of the surface modifiers - their removal via DMF washing reduced the polar component of surface energy. Consequently, the decreased surface modifier content diminished the substrates' hydrophilicity, manifesting as increased contact angles.

Contact angle measurements were performed using ultrapure water droplets as the test liquid. Figure S15 A-D shows the contact angles of different SnO₂ substrates: (A) pure SnO₂, (B) SnO₂-DTPA, (C) SnO₂/Zol, and (D) SnO₂-DTPA/Zol. All modified SnO₂ substrates demonstrated better wettability than pure SnO₂. The substrates were washed with large amounts of DMF. Specifically, 1000 μL DMF was dropped onto different SnO₂ substrates, allowed to stand for about 15 seconds for complete wetting, then spin-coated at 3000 rpm for 30 seconds, and heated at 150°C for 5 minutes to remove the solvent.

Figure S15 E-H shows the corresponding contact angles of SnO₂ substrates after extensive DMF washing: (E) pure SnO₂, (F) SnO₂-DTPA, (G) SnO₂/Zol, and (H) SnO₂-DTPA/Zol. After extensive DMF washing: The contact angle of pure SnO₂ substrate showed almost no change, proving that DMF washing treatment has little effect on the wettability of SnO₂ itself. The contact angles of SnO₂-DTPA and SnO₂/Zol substrates increased significantly, indicating partial desorption of DTPA and Zol molecules from the SnO₂ surface. The contact angle of the SnO₂-DTPA/Zol sample showed almost no change and still exhibited good wettability. This proves that co-modification with DTPA and Zol can ensure that SnO₂ does not fail due to desorption under the load of upper film solvents.

4# In Figure S10, why did the N 1s and P 2p peaks disappear for the SnO₂-DTPA sample?

Our Response: Thank you very much for this comment. We have supplemented the relevant content in the Supporting Information (Line 79-83, Page 12). Since DTPA molecules are uniformly

dispersed throughout the entire SnO₂ layer with minimal additive amounts, and considering that XPS can only detect the chemical environment at the sample surface with limited instrument sensitivity, we were unable to detect trace amounts of DTPA molecules on the SnO₂-DTPA sample surface. This explains the absence of N 1s and P 2p peaks. In contrast, the Zol modifier is entirely concentrated on the SnO₂ surface with relatively higher concentration, which enables clear detection of distinct N 1s and P 2p peaks.

5# Figure 2D presents the differential charge density, but the authors did not discuss this in the main text. It is unclear what specific result or insight the authors aim to convey through this figure.

Our Response: Thank you very much for this comment. As you mentioned, the discussion on the differential charge density in Figure 2D was insufficient in the manuscript. We have supplemented the relevant content in the main text (Line 171, Page 8). The specific additions are copied below:

The differential charge density in Figure 2D show significant charge transfer between the two modifier molecules (DTPA and Zol) and SnO₂. The corresponding electron localization function (ELF) images demonstrate electron cloud overlap between Sn and O atoms (Figure R2 and Figure S11), further confirming the strong coupling effect between the groups (-COOH in DTPA, -PO(OH)₂ in Zol) and the SnO₂ lattice. These results prove the strong interactions between our selected modifier molecules and the SnO₂ substrate.

Figure R2. The ELF image of A) DTPA and B) Zol absorption on SnO₂ surface, obtained through DFT calculations.

6# In Figures 2F-I, the -OH peak area for the pure SnO₂ sample after washing reaches ~10.5%, which is higher than that of the SnO₂-DTPA and SnO₂/Zol samples. This appears to be unusual and warrants further explanation.

Our Response: Thank you very much for this comment. We have supplemented the relevant content in the Supporting Information (Line 124-140, Page 16). The formation of -OH groups on pure SnO₂ surfaces primarily occurs through two pathways:

1) Dissociation of adsorbed water molecules: When exposed to air, SnO₂ surfaces physically adsorb H₂O molecules, which may subsequently dissociate to form chemically adsorbed -OH and H⁺ ($\text{H}_2\text{O} + \text{Sn-O-Sn} \rightarrow \text{Sn-OH} + \text{Sn-OH}$).

2) Reaction of surface oxygen vacancies with water/oxygen: Oxygen vacancies (V_o) generated during SnO₂ film preparation can react with environmental H₂O or O₂ to form hydroxyl groups: a) $\text{V}_o + \text{H}_2\text{O} \rightarrow 2\text{OH}^-$; b) $\text{V}_o + 1/2\text{O}_2 + \text{H}_2\text{O} \rightarrow 2\text{OH}^-$. Generally, more oxygen vacancies lead to higher surface hydroxylation.

In Figure 2F, the -OH peak area of pure SnO₂ remains nearly unchanged after DMF rinsing, indicating its surface chemical state is unaffected by DMF treatment.

In Figures 2G and H, SnO₂-DTPA and SnO₂/Zol samples inherently contain more -OH groups than pure SnO₂ sample. This is because both DTPA and Zol molecules contain abundant -COOH and -PO(OH)₂ groups. During sample preparation, some of these -OH groups react and bond with the SnO₂ surface, while the remaining unbonded -OH groups increase the degree of surface hydroxylation.

On the other hand, the modification of SnO₂ by -COOH and -PO(OH)₂ reduces the intrinsic defect concentration on the SnO₂ surface. Consequently, after extensive DMF washing, some DTPA and Zol molecules are removed, leading to a decrease in -OH groups. The exposed defects, now fewer in number, undergo minimal hydroxylation. As a result, the -OH peak area of the washed SnO₂-DTPA and SnO₂/Zol samples becomes smaller than that of the pure SnO₂ sample.

(References: DOI: 10.1016/j.progsurf.2005.09.002; DOI: 10.1016/S0167-5729(01)00020-6; DOI: 10.1016/S0167-5729(02)00100-0; DOI: 10.1002/adv.202300010; DOI: 10.1002/adma.202110438; DOI: 10.1002/aenm.202400791)

7# Similarly, the DFT results presented in Figures 3B-E and the NMR test results in Figures S13-S14 require more detailed descriptions and thorough discussion to fully support the findings. If the authors are constrained by space limitations, they may consider adding these details to the Supporting Information (SI).

Our Response: Thank you very much for this comment. We have supplemented the corresponding DFT and NMR analyses in the manuscript (line 218, Page 11) and in the Supporting Information (Page 19, Page 20).

Figures 3A and B respectively depict the significant differences in charge density between PVK with unsaturated lattice and saturated lattice interacting with Zol molecules. The changes in electron cloud distribution indicate charge gain and loss, with red and blue regions representing electron depletion and accumulation, respectively, due to electron redistribution. In Figure 3A, the electron density increases on the FA⁺ surface while decreasing on the N atom of the Zol molecule

above it, suggesting that Zol bonds with FA^+ through the N atom, thereby transferring electrons to FA^+ . Similarly, in Figure 3B, the electron density increases on the Pb atom surface but decreases on the N atom surface, indicating that Zol can also bind to Pb via the N atom, leading to electron transfer to Pb. This is further reflected in the differential charge density analysis (Figure 3C), where the PVK surface gains electrons, forming a negative charge center, while Zol loses electrons, creating a positive charge center. Consequently, an efficient carrier transfer is established at the Zol/PVK interface.

^1H NMR spectroscopy was employed to investigate the interaction between Zol and FAI (Figure S17). Upon modification of Zol, the hydrogen peak of FAI's $-\text{NH}_2$ group at 8.83 ppm split into two new peaks at 9.01 ppm and 8.66 ppm. Additionally, the two $-\text{NH}_2$ hydrogen peaks of FAPbI_3 , originally at 8.93 ppm and 8.74 ppm respectively, shifted to 9.01 ppm and 8.66 ppm following Zol modification. As consistently demonstrated in Figures 3D and E, these observations confirm the formation of robust $\text{N-H}\cdots\text{N}$ hydrogen bonding interactions between Zol and FAI.

^1H -NMR was used to study the interaction between Zol and PbI_2 (Figure S18). Compared with pure Zol, shift of δ values were observed when Zol was mixed with PbI_2 . The hydrogen peak of Zol at 8.87 ppm shifted to 8.84 ppm, the peak at 7.60 ppm split into two peaks, and the peak at 7.47 ppm disappeared. This is attributed to the interaction between Zol and PbI_2 distorting the electron density cloud around the hydrogen nuclei, which affects the magnetic field changes and consequently alters the shielding effect on the hydrogen atoms.

8# The authors claim that the XRD peak intensity increases after modification, but they did not provide a clear conclusion regarding what this intensity enhancement signifies. It is important to note that peak intensity can be influenced by multiple factors. Additionally, it remains unclear why the modification is beneficial for the crystallization and growth of the FAPbI_3 film. Further explanation is needed to clarify these points.

Our Response: Thank you very much for this comment. We have supplemented the relevant content in the manuscript (Line 243, Page 13).

The XRD spectrum showed that the characteristic peaks of the perovskite film on the SnO_2 electrode before and after modification were essentially the same, while the peak intensity of the perovskite film on the SnO_2 -DTPA/Zol electrode was significantly enhanced (Figure 3F). This indicates that the atomic arrangement orderliness at the lower interface of the modified perovskite film has been significantly improved.

Through combined theoretical calculations and ^1H -NMR spectroscopy, we have demonstrated that the modifier interacts with FA^+ in perovskite to form robust $\text{N-H}\cdots\text{N}$ hydrogen bonds. Moreover, the modifier also interacts with Pb in the perovskite. We therefore propose a dual mechanism: During PbI_2 deposition, Zol can bind with Pb, promoting ordered arrangement at the bottom PbI_2 layer. Simultaneously, during FAI deposition, Zol molecules on the substrate surface rapidly coordinate with FA^+ , resulting in a more ordered terminal layer at the perovskite bottom with reduced defects and enhanced crystallinity. This explains the observed XRD peak intensity enhancement. The increased XRD intensity indicates improved crystal perfection with reduced terminal defect density, which facilitates charge extraction and suppresses non-radiative

recombination. These modifications are fundamentally beneficial for both photovoltaic performance and stability of PSC devices. The relevant analyses have been supplemented in the Supporting Information (Line 185-188, Page 20).

9# *Although the authors confirmed the occurrence of chemical crosslinking between DTPA and Tol molecules through extensive spectroscopic analysis, there is a lack of visual evidence (e.g., TEM, DOI: 10.1002/adma.202300403; DOI: 10.1038/s41467-018-06204-2) to demonstrate whether this linking penetrate the entire oxide transport layer.*

Our Response: Thank you very much for this comment. We sincerely appreciate your valuable suggestions. The TEM-EDS characterization has been supplemented (Line 212, Page 11 in the manuscript; Figures R3 and S16):

Cross-sectional TEM with EDS mapping was employed to probe the spatial distribution of both DTPA and Zol additives throughout the thickness (Figures R3 and S16). As revealed by the EDS profiles, P and C elements exhibit distinct and homogeneous distribution within the SnO₂ layer, demonstrating uniform dispersion of the modifiers across the entire SnO₂ transport layer (Figures R3 and S16).

Figure R3. Cross-sectional HAADF and corresponding TEM-EDS image of glass/FTO/SnO₂-DTPA/Zol/PVK.

10# *In Figure 4E, why did the PL peak of the SnO₂-DTPA-perovskite sample exhibit a red shift compared to the other samples?*

Our Response: Thank you very much for this comment. After a lot of discussion and analysis, we attributed this red shift to phase separation on the surface of the samples due to the aging of the sample. Considering this situation, we re-prepared the sample and characterized it. Figure R4 demonstrates consistent results across all three replicates, demonstrating that none of the samples displayed any peak red shift. We have corrected this deviation in Figure 4E of the manuscript.

Figure R4. Three reproductions of PL testing: A) First reproduction, B) Second reproduction, C) Third reproduction.

11# In Figure 4F, the TRPL curves normalized? Why is the fitted parameter A_1 greater than 1?

Our Response: Thank you very much for this comment. We initially performed fitting on data normalized using Origin software, but the results were found to be erroneous. Consequently, we re-analyzed the raw, non-normalized data (Figure R5 and Table R4). The corrected results have been updated in both the manuscript and Supporting Information (Figure 4F and Table S9). Additionally, we have included detailed fitting methodology in the Supporting Information (Line 219, Page 26):

The PL decay fitting curve is based on the bi-exponential decay equation:

$$f(t) = A_1 \exp\left(\frac{-t}{\tau_1}\right) + A_2 \exp\left(\frac{-t}{\tau_2}\right) + y_0$$

Where A_1 and A_2 represent the decay amplitude, τ_1 represents trap-assisted recombination, τ_2 represents free carrier recombination, and y_0 is a constant for baseline offset.

Figure R5. TRPL of FAPbI₃ films grown on different SnO₂ electrodes before and after modification.

Table R4. Fitted TRPL parameters for perovskite films with different ETLs.

Sample	A ₁	A ₂	τ ₁ (ns)	τ ₂ (ns)	τ _{ave} (ns)
FTO/SnO ₂ /Perovskite	1086.320	589.780	54.25	354.69	159.97
FTO/SnO ₂ -DTPA/Perovskite	248.050	257.890	41.46	246.01	145.72
FTO/SnO ₂ /Zol/Perovskite	519.560	310.800	32.11	252.43	114.57
FTO/SnO ₂ -DTPA/Zol/Perovskite	391.540	237.590	24.48	199.22	90.47

12# The authors used XRD to test stability of films (Figures S21 and S22), why did the PbI₂ peaks for some samples exhibit a shift?

Our Response: Thank you very much for this comment. For the original Figure S21 (now Figure R6 and S26): The shift in peak positions within the 11-13° range before and after aging is attributed to the phase transition of perovskite from the α-phase to δ-phase. We have refined the labeling of PbI₂ and δ-phase peak positions in the XRD patterns (in Figures R6 and S26).

Figure R6. The XRD patterns of UV light aging experiments for perovskite films grown on different SnO₂ substrates. (254nm UV, T_{ambient}=35°C, RH=85%).

For the original Figure S22 (now Figure R7): all PbI₂ peaks appeared within the range of 12.55-12.60° with variations less than 0.05°. We therefore attribute these minor differences to instrumental errors during XRD measurements rather than peak shifts caused by modifiers or other factors.

In addition, to further verify whether the characteristic PbI₂ peaks in the original Figure S22 (now Figure R7) would indeed shift, we reproduced the original experimental conditions (110°C annealing, T_{ambient}=35°C, RH=85%, natural lighting) and repeated the damp heat test for perovskite films grown on different SnO₂ substrates, followed by additional XRD characterization. As shown in Figure R8 and S27, no peak shift was observed for PbI₂ in any sample. Besides, we observed a left-shift of the PVK characteristic peaks after 17 hours of aging, likely due to slight thermal-induced lattice distortion in the perovskite crystals. The intensified PbI₂ peaks after aging further confirmed the collapse of the perovskite lattice, where FA⁺ escaped and transformed the perovskite into PbI₂. We have corrected this deviation in Figure S27 of the SI.

Figure R7. The XRD patterns of the damp heat test for perovskite films grown on different SnO₂ substrates. (110 °C Annealing, T_{ambient}=35°C, RH=85%, Natural Lighting).

Figure R8. Reproduction of the XRD testing: the damp heat experiments for perovskite films grown on different SnO₂ substrates. (110 °C Annealing, T_{ambient}=35°C, RH=85%, Natural Lighting)

13# Why did the stability tests in Figures 5C, E, and G not display the initial values? The initial efficiency of the devices should be clearly indicated.

Our Response: Thank you very much for this comment. We replaced all data with absolute PCE values instead of normalized data to more intuitively reflect the device stability (Figures 5A, 5C, 5E). Additionally, for Figure 5G, we have clearly labeled the initial PCE of the devices before 1000 h damp heat pre-exposure in the figure.

14# The authors used Spiro-OMeTAD as the hole transport layer (HTL) in PSCs for both PCE and stability tests. It should be noted that Li-doped Spiro-OMeTAD is inherently unstable, particularly under humidity, light, and thermal conditions. Some of the important references related to this point are recommended for citation and discussion (DOI: 10.1126/science.abo27; DOI: 10.1038/s41467-024-52199-4; DOI: 10.1002/anie.202316183).

Our Response: Thank you very much for this comment. We sincerely appreciate your recommendation of these important references, which we have now cited in our manuscript. Regarding your concern about the intrinsic instability of Li-doped Spiro-OMeTAD, we actually incorporated PTAA into the Spiro-OMeTAD and modified the upper surface of the perovskite films to ensure good stability for our stability test samples. We have therefore added detailed preparation methods for these stability test samples in the Experimental section (Line 422-442, Page 21-22):

The stability analysis devices:

Under heating conditions, PEAI could be gradually converted into the PEA_2PbI_4 phase, leading to reduced device operational lifetime³⁹. Consequently, we removed the PEAI passivation layer for

all thermal-related stability tests. Additionally, given the intrinsic instability of Spiro-OMeTAD as a HTL⁴⁰⁻⁴², we adopted a Spiro-OMeTAD/PTAA hybrid HTL for most stability evaluations, while employing pure PTAA HTL for a limited number of tests to ensure enhanced stability.

Specifically, for thermal stability testing, the perovskite films received no surface modification with PEAI or any other agent. The Spiro-OMeTAD/PTAA mixed solution (90/15 mg/mL in chlorobenzene) was supplemented with 35 μ L Li-TFSI solution (260 mg/mL in acetonitrile), 30 μ L tBP, and 30 μ L FK209 Co(III) TFSI salt (375 mg/mL in acetonitrile), followed by spin-coating deposition at 5000 rpm for 30 seconds.

For storage, MPPT, and light-dark cycling tests, the perovskite films were modified with OAI (5 mg in 1 mL isopropanol) through spin-coating at 4000 rpm for 30 seconds and subsequently annealed at 150°C for 10 minutes to form a 2D perovskite surface layer. The Spiro-OMeTAD/PTAA mixed solution (90/7 mg/mL in chlorobenzene) was supplemented with 35 μ L Li-TFSI solution (260 mg/mL in acetonitrile), 30 μ L tBP, and 30 μ L FK209 Co(III) TFSI salt (375 mg/mL in acetonitrile), and deposited via spin-coating at 5000 rpm for 30 seconds.

Regarding thermal cycling and outdoor storage stability tests, the perovskite films remained unmodified with PEAI or other surface treatments. The PTAA solution (20 mg/mL in chlorobenzene) was mixed with 10 μ L tBP and 2.6 μ L Li-TFSI solution (260 mg/mL in acetonitrile), then spin-coated at 5000 rpm for 30 seconds.

15# The authors emphasize the importance of dynamic stability testing, but the manuscript only provides light cycling stability data. Additional dynamic stability tests, such as thermal cycling and even real outdoor stability, should be included to comprehensively evaluate the performance.

Our Response: Thank you very much for this comment. We have conducted additional dynamic stability tests (Line 345, Page 17, Figures R9 and S31, Figures R10 and S32).

Given the inherent instability of Spiro-OMeTAD as a hole transport layer (HTL), **we employed pure PTAA as the HTL** to ensure higher device stability. The specific fabrication method is as follows (Lines 439-442, Page 22 in the manuscript): Regarding thermal cycling and outdoor storage stability tests, the perovskite films remained unmodified with PEAI or other surface treatments. The PTAA solution (20 mg/mL in chlorobenzene) was mixed with 10 μ L tBP and 2.6 μ L Li-TFSI solution (260 mg/mL in acetonitrile), then spin-coated at 5000 rpm for 30 seconds.

Thermal cycling experiments were performed on devices with an active area of 0.04 cm² (Figures R9 and S31). Each cycle consisted of: (1) aging the devices in a vacuum drying oven at 65°C for 24 hours, followed by (2) storage in a N₂ glovebox under dark conditions at room temperature for 24 hours. After 960 h, the unencapsulated target devices retained over 80% of their initial PCE, while the control devices degraded to below 40% of their initial PCE.

Figure R9. Efficiency evolution of PSCs under thermal cycling tests (24 h on/24 h off)

Real outdoor stability testing was also performed on 0.04 cm² devices (Figures R10 and S32). The encapsulated devices were aged under actual outdoor conditions. After 500 hours, the target devices maintained approximately 80% of their initial PCE, whereas the control devices degraded to less than 20% of their initial PCE.

Figure R10. Efficiency evolution of PSCs under real outdoor storage tests

16# In the Experimental Section, more details are necessary regarding certain preparation and testing procedures. For example, the authors claim that the perovskite was prepared in air, but specific details are not provided. Additionally, more information should be included about the stability testing protocols to ensure reproducibility and clarity.

Our Response: Thank you very much for this comment. We have incorporated the specific experimental and testing details you suggested in the manuscript (Page 20-22, line 394-442):

Device fabrication:

SnO₂ Substrate Preparation: The FTO glass was successively cleaned in detergent, deionized water, and ethanol for 30 min by an ultrasonic cleaner. The clean FTO was treated with an ultraviolet ozone machine for 30 min. The SnO₂ solution was mixed with water at a volume ratio

of 1:3 and spin-coated on the FTO at 3000 rpm for 30 seconds, followed by annealing at 150 °C for 30 min in air.

PSCs Fabrication: The whole process of the PSCs fabrication was carried out in an ambient air environment (room temperature and 20-50% relative humidity). Perovskite films were fabricated by either one-step spin-coating or two-step spin-coating. For surface passivation, 4 mg/mL PEAI in isopropanol was spin-coated on the perovskite film at 5000 rpm for 30 s without further treatment. The hole transfer material (HTM) solution was prepared by mixing 90 mg Spiro-OMeTAD, 35 μ l bis(trifluoromethane) sulfonimide lithium salt (Li-TFSI) solution (260 mg Li-TFSI in 1 ml acetonitrile), 30 μ l 4-tertbutylpyridine (tBP), and 30 μ l of FK209 Co(III) TFSI salt (375 mg/mL in acetonitrile). Then, the HTM solution was deposited on the top at 3000 rpm for 30 s in air without annealing. Finally, 80 nm Au was deposited to form an electrode by thermal evaporation under 5×10^{-4} Pa. Before the J-V test, a commercial anti-reflection layer (Shengyu Tech.) is carefully stuck on the glass side for better light transmittance.

(1) The perovskite films which were fabricated by the two-step solution process:

Firstly, 692 mg PbI_2 was dissolved in 1 ml DMF: DMSO (v: v=9:1), spin-coated on the SnO_2 electrode at 1500 rpm for 30 seconds, and annealed at 70 °C for 1 minute. After cooling down to room temperature, the organic solution (92.5mg FAI, 6.5mg MAI, and 9.5mg MAI in 1mL IPA, stirred for more than 2 h) was spin-coated onto the PbI_2 film at 2500 rpm for 3 s, 1350 rpm for 15 s and 1700 rpm for 12 s. The film was annealed at 150 °C for 15 min. All spin-coating and annealing processes were performed under ambient air conditions (room temperature and 20-50% R.H.).

(2) The perovskite films which were fabricated by the one-step solution process³⁸:

The perovskite precursor solution was prepared by dissolving 810 mg of black-phase FAPbI_3 , 35 mol% of MAI, and 1 mol% of acetylcholine chloride in DMF and DMSO (8:1 v/v) at 60 °C. The perovskite layers were then spin-coated at 1,000 rpm for 10 s and 5,000 rpm for 15 s, and 1 mL of ethyl ether was dripped onto the electrode during spin-coating. The perovskite layers were annealed at 120 °C for 40 min, and the electrode was cooled. All spin-coating and annealing processes were performed under ambient air conditions (room temperature and 20-50% R.H.).

(3) The stability analysis devices:

Under heating conditions, PEAI could be gradually converted into the PEA_2PbI_4 phase, leading to reduced device operational lifetime³⁹. Consequently, we removed the PEAI passivation layer for all thermal-related stability tests. Additionally, given the intrinsic instability of Spiro-OMeTAD as a HTL⁴⁰⁻⁴², we adopted a Spiro-OMeTAD/PTAA hybrid HTL for most stability evaluations, while employing pure PTAA HTL for a limited number of tests to ensure enhanced stability.

Specifically, for thermal stability testing, the perovskite films received no surface modification with PEAI or any other agent. The Spiro-OMeTAD/PTAA mixed solution (90/15 mg/mL in chlorobenzene) was supplemented with 35 μ L Li-TFSI solution (260 mg/mL in acetonitrile), 30

μL tBP, and 30 μL FK209 Co(III) TFSI salt (375 mg/mL in acetonitrile), followed by spin-coating deposition at 5000 rpm for 30 seconds.

For storage, MPPT, and light-dark cycling tests, the perovskite films were modified with OAI (5 mg in 1 mL isopropanol) through spin-coating at 4000 rpm for 30 seconds and subsequently annealed at 150°C for 10 minutes to form a 2D perovskite surface layer. The Spiro-OMeTAD/PTAA mixed solution (90/7 mg/mL in chlorobenzene) was supplemented with 35 μL Li-TFSI solution (260 mg/mL in acetonitrile), 30 μL tBP, and 30 μL FK209 Co(III) TFSI salt (375 mg/mL in acetonitrile), and deposited via spin-coating at 5000 rpm for 30 seconds.

Regarding thermal cycling and outdoor storage stability tests, the perovskite films remained unmodified with PEAI or other surface treatments. The PTAA solution (20 mg/mL in chlorobenzene) was mixed with 10 μL tBP and 2.6 μL Li-TFSI solution (260 mg/mL in acetonitrile), then spin-coated at 5000 rpm for 30 seconds.

Reviewer #2

In this manuscript, Li et. al. reports a fully chemical modification strategy in which the interfacial modifiers undergo an in-situ crosslinking-like reaction, forming a localized, chemically bonded layer that seamlessly extends from the bulk of the underlying transport layer to the interface. Perovskite solar cells (PSCs) fabricated with this fully chemical modification strategy achieved a power conversion efficiency (PCE) of 25.52% under standard conditions, representing a high PCE for devices fully fabricated in an ambient atmosphere. In addition, these cells also show excellent stability. This work is interesting and well structured. However, some issues should be addressed before further consideration. Here are the specific comments:

Our Response: We thank to the reviewer very much for overall positive comments.

1# The author emphasizes that the use of DTPA and Zol as modifiers can significantly enhance the extraction and migration of interfacial charges. However, the manuscript lacks direct experimental evidence for the kinetic recombination process of photogenerated carriers at the bottom of the perovskite. It is recommended to supplement with femtosecond time-resolved transient absorption spectroscopy (fs-TA) to provide more direct real-time evidence.

Our Response: Thank you very much for this comment. We have supplemented femtosecond transient absorption spectroscopy (fs-TAS) measurements (Line 199, Page 10 in the manuscript and Figure S14), with the specific results and analyses as follows:

Femtosecond transient absorption spectroscopy (fs-TAS) was employed to investigate the charge carrier dynamics between modified SnO₂ and PVK. In both cases, a strong and immediate ground-state bleaching (GSB) peak at approximately 784 nm was observed upon excitation of the host perovskite absorption (Figure R11 A and B). Typically, the GSB signal is proportional to carrier density. For the SnO₂/perovskite stack, the variation in GSB peak intensity with delay time directly reflects the quantity of photogenerated carriers in the conduction and valence bands of the perovskite²⁷. Compared to pristine SnO₂/PVK, weaker GSB signals at identical delay times were exhibited by the SnO₂-DTPA/Zol/PVK sample (Figure R11 C and D), along with a higher electron extraction rate (Figure R11 E). It is concluded that faster electron transfer from the perovskite to the adjacent modified SnO₂ contact occurs. This can be attributed to the critical role of interface modification in promoting carrier extraction.

Figure R11. Femtosecond transient absorption spectroscopy (fs-TAS) of (A) SnO₂/PVK and (B) SnO₂-DTPA/Zol/PVK films. Corresponding TA spectra at different decay times of (C) SnO₂/PVK and (D) SnO₂-DTPA/Zol/PVK films. (E) Corresponding TA decay kinetics.

2# This study confirms that the in-situ crosslinking-like reaction enabled by the two modifiers, DTPA and Zol, can significantly prevent the desorption of the modifiers. However, the mechanistic explanation for this phenomenon is not yet entirely clear. Please clarify it.

Our Response: Thank you very much for this comment. The relevant analyses have been supplemented in the Supporting Information (Line 51-56, Page 5). Conventional surface modifiers or interface optimization molecules typically interact with SnO₂ through only one or a few binding sites, which inherently limits their overall adsorption strength. In contrast, our DTPA and Zol modifiers undergo in situ crosslinking-like reactions that enable multiple DTPA molecules to form an extended network structure through Zol bridging. The interconnected molecular framework enables cooperative interactions among adjacent binding sites, thereby dramatically enhancing the overall adsorption capability of the modifier system.

3# In Fig. 3F, the characteristic peaks of PbI₂ in the XRD pattern are high, which seems inconsistent with high efficiency PSCs devices. Please confirm it.

Our Response: Thank you very much for this comment. In the fabrication of perovskite films via the full air process, real-time experimental conditions significantly impact the crystallinity quality of the films. Therefore, we sincerely apologize for the suboptimal performance of the previously prepared batch of perovskite films due to experimental condition limitations. We have repeated the experiments and prepared a new batch of higher-quality films, followed by additional XRD characterization. As shown in Figures R12 and 3F, the observed trends remain consistent with

previous findings. After co-modification of the substrate with DTPA and Zol, the crystallinity of the deposited perovskite films showed remarkable improvement.

Figure R12. XRD patterns of FAPbI₃ films grown on different SnO₂ electrodes before and after modification.

4# In Fig. S21 and S22, the XRD pattern have been provided to analyze the phase stability of FAPbI₃ films. However, more evidence such as the SEM image of different samples should also be given to further prove the improvement of stability of perovskite films.

Our Response: Thank you very much for this comment. We conducted SEM characterization of the films before and after aging tests (Figures R13 and S28). Figures R13 A-D show SEM images of four samples before aging. Figures R13 E-H present SEM images of samples after thermal aging at 110°C for 17 hours. The pure SnO₂ substrate samples exhibited numerous large pinholes, while the SnO₂-DTPA/Zol substrate samples showed significantly fewer pinholes. Figures R13 I-L display SEM images of samples after UV light aging for 17 hours. The perovskite films on pure SnO₂ substrates showed completely collapsed grain structures with extensive pinholes and cracks, whereas films on SnO₂-DTPA/Zol substrates maintained relatively intact perovskite grain structures with only slight surface decomposition and no large pinholes or cracks.

Additionally, we performed absorption spectroscopy measurements on samples after 17 hours of UV exposure. As shown in Figures R14 and S29, the pure SnO₂ substrate samples demonstrated substantially reduced light absorption capability between 500-900 nm after aging, while the SnO₂-DTPA/Zol substrate samples maintained relatively strong light absorption capacity even after 17 hours of aging.

Figure R13. The SEM characterization of the films before and after aging tests.

Figure R14. The UV-vis spectroscopy measurements on samples after 17 hours of UV exposure.

5# The authors mention the concepts of “statically and dynamically stable” in the title and on lines 282-284 of the manuscript. It is recommended to describe this concept in the Introduction section so that readers can better understand it.

Our Response: Thank you very much for this comment. We have supplemented the description of this concept in the Introduction section (Line 72, Page 4):

Developing failure-resistant interface modification methods is crucial for enhancing device stability under both static conditions (constant environmental parameters such as fixed temperature, humidity, or continuous illumination) and dynamic conditions (simulating real-world scenarios including thermal cycling, light-dark cycling, or actual outdoor exposure).

6# As far as we know, PEAI modification alone seems insufficient to achieve such remarkable stability. Did the authors use any other modifiers during the MPPT and thermal stability tests? If so, please supplement detailed descriptions so that readers can reproduce the experiment.

Our Response: Thank you very much for this comment. We have incorporated the specific experimental and testing details you suggested into the Experimental section (Line 422-442, Page 21-22):

The stability analysis devices:

Under heating conditions, PEAI could be gradually converted into the PEA_2PbI_4 phase, leading to reduced device operational lifetime³⁹. Consequently, we removed the PEAI passivation layer for all thermal-related stability tests. Additionally, given the intrinsic instability of Spiro-OMeTAD as a HTL⁴⁰⁻⁴², we adopted a Spiro-OMeTAD/PTAA hybrid HTL for most stability evaluations, while employing pure PTAA HTL for a limited number of tests to ensure enhanced stability.

Specifically, for thermal stability testing, the perovskite films received no surface modification with PEAI or any other agent. The Spiro-OMeTAD/PTAA mixed solution (90/15 mg/mL in chlorobenzene) was supplemented with 35 μL Li-TFSI solution (260 mg/mL in acetonitrile), 30 μL tBP, and 30 μL FK209 Co(III) TFSI salt (375 mg/mL in acetonitrile), followed by spin-coating deposition at 5000 rpm for 30 seconds.

For storage, MPPT, and light-dark cycling tests, the perovskite films were modified with OAI (5 mg in 1 mL isopropanol) through spin-coating at 4000 rpm for 30 seconds and subsequently annealed at 150°C for 10 minutes to form a 2D perovskite surface layer. The Spiro-OMeTAD/PTAA mixed solution (90/7 mg/mL in chlorobenzene) was supplemented with 35 μL Li-TFSI solution (260 mg/mL in acetonitrile), 30 μL tBP, and 30 μL FK209 Co(III) TFSI salt (375 mg/mL in acetonitrile), and deposited via spin-coating at 5000 rpm for 30 seconds.

Regarding thermal cycling and outdoor storage stability tests, the perovskite films remained unmodified with PEAI or other surface treatments. The PTAA solution (20 mg/mL in chlorobenzene) was mixed with 10 μL tBP and 2.6 μL Li-TFSI solution (260 mg/mL in acetonitrile), then spin-coated at 5000 rpm for 30 seconds.

Reviewer #3

In the manuscript titled “Fully Chemical Interface Engineering for Statically and Dynamically Stable Perovskite Solar Cells,” Nie et al. introduced a chemical adsorption strategy to modify the interface between SnO₂ and perovskite, achieving a power conversion efficiency (PCE) of 25.52%. They also presented comprehensive stability studies demonstrating the durability of the devices. However, I believe the manuscript requires significant improvements before it can be reconsidered for publication in Nature Communications.

Our Response: We thank to the reviewer very much for overall positive comments.

First, the reported efficiency has not been certified. It is now standard practice to certify the efficiency of perovskite solar cells, especially given the numerous measurement uncertainties associated with them. Moreover, it is worth noting that the in-house PCE claimed is significantly below the record PCE of 27% for perovskite solar cells.

Our Response: Thank you very much for this comment. In consideration of your concerns regarding device performance, we provide the following explanation:

The devices fabricated in fully air environment, with specific experimental details supplemented in the Experimental section of the manuscript (Line 394-421, Page 20-21). The PCE of our PSCs has been certified by a third party, with the certification results presented in Figures R15 and S20 (PCE=25.49%, The device was encapsulated and equipped with an AR film). Additionally, we also compared the efficiency of recently reported high-performance devices prepared in fully air environments (Table S8 and R5), demonstrating the advancement of our devices.

Figure R15. Device certification results in NIMTT.

- Report No.:** 检测字第 202505100834 号
- 报告日期:** 2025 年 05 月 29 日
- 样品名称:** Perovskite solar cells
- 标称生产单位:** Nanjing University of Aeronautics and Astronautics
- 委托单位:** Nanjing University of Aeronautics and Astronautics
- 联系地址:** No. 29, Yudao Street, Qinhuai District, Nanjing, Jiangsu Province, China
- 检测类别:** Commission test

检测结果

Results of Test

1. Test Condition

Reference Cell: Mono-Si solar cell (window material: KG1).

Sample Information: Perovskite Solar Cells. Dimensions:(2.5×2.5) cm². Aperture area:0.0969 cm².

Storage Condition of Sample Before Test: Temperature: 25±5℃; humidity: 30±10%; stored in dark for 3 Days.

2. Methodologies and Settings

(1) In the 3A steady-state solar simulator (spectrum: AM1.5), the irradiance of the solar simulator was first calibrated to 1000 W/m² with a standard solar cell, the temperature of the solar cell was controlled by a water bath thermostat at 25℃, and then the I-V parameters of the tested sample were measured with a digital source meter.

(2) The parameter Settings of the measurement software are shown in Table 1:

Table 1 Parameter Settings for I-V Test

Scan Mode	Start Voltage(V)	End Voltage(V)	Step (V)	Sweep point delay(ms)	Number of point	Light Soaking Pre-treatment
Reverse scan	1.3	-0.01	0.01	50	141	YES

3. Test Results

Current-voltage (I-V) curves are shown in Figures 1 and Table 2:

Figure 1 I-V Curve (Reverse Scan)

Table 2 I-V Parameter

Scan Mode	I_{sc} (mA)	V_{oc} (V)	FF (%)	P_{max} (mW)	V_{pmax} (V)	I_{pmax} (mA)	η (%)
Reverse scan	2.53	1.18	82.75	2.47	1.02	2.42	25.49

备注
Note

- Reported performance parameters take from one test value.
- The solar cell area data is determined according to the mask area.

审核人员
Verified by

吴伟钢

主检人员
Tested by

康张李

Table R5. Summary of PCEs of high-efficiency (PCE > 24.5%) devices prepared in a full-air environment.

Perovskite	Relative Humidity (%)	Champion PCE (%)	Device structure	References
FAPbI ₃	20-50	25.52 (certificated 25.49)	n-i-p	This Work
FAPbI ₃	20	24.7	n-i-p	Y. Zou, Science, 2024 ¹
FAPbI ₃	30-85	25.74 (certificated 25.43)	n-i-p	Y. Yang, Adv. Energy Mater. 2024 ²
CS _{0.01} (FA _{0.97} MA _{0.03}) _{0.99} Pb(I _{0.97} Br _{0.03}) ₃	35-50	24.72	p-i-n	H. Meng, Nature Energy, 2024 ³

Second, there seems to be a considerable amount of missing information in the experimental section. For instance, the authors claimed that the devices demonstrated stability at 85°C. This is questionable, as the use of the Spiro-MeOTAD hole transport layer in this device structure contradicts this claim; many studies have reported that doped Spiro-MeOTAD degrades at elevated temperatures. Additionally, in Figure 5, the authors presented normalized PCE instead of absolute PCE, which can be misleading. Devices with moderate performance may be able to improve their performance during aging tests, leading to an inaccurate representation

Our Response: Thank you very much for this comment. We have supplemented the experimental details regarding the preparation of stability test samples in the Experimental section. In response to your concerns about the inherent instability of the Spiro-OMeTAD hole transport layer, we incorporated PTAA into Spiro-OMeTAD during the fabrication of stability test samples due to PTAA's superior thermal stability. Furthermore, since PEAI is also prone to thermal decomposition, we eliminated the PEAI modification on the upper surface of the perovskite film to ensure optimal device stability under high-temperature conditions. The specific experimental protocols are as follows (Line 422-442, Page 21-22):

The stability analysis devices:

Under heating conditions, PEAI could be gradually converted into the PEA_2PbI_4 phase, leading to reduced device operational lifetime³⁹. Consequently, we removed the PEAI passivation layer for all thermal-related stability tests. Additionally, given the intrinsic instability of Spiro-OMeTAD as a HTL⁴⁰⁻⁴², we adopted a Spiro-OMeTAD/PTAA hybrid HTL for most stability evaluations, while employing pure PTAA HTL for a limited number of tests to ensure enhanced stability.

Specifically, for thermal stability testing, the perovskite films received no surface modification with PEAI or any other agent. The Spiro-OMeTAD/PTAA mixed solution (90/15 mg/mL in chlorobenzene) was supplemented with 35 μL Li-TFSI solution (260 mg/mL in acetonitrile), 30 μL tBP, and 30 μL FK209 Co(III) TFSI salt (375 mg/mL in acetonitrile), followed by spin-coating deposition at 5000 rpm for 30 seconds.

For storage, MPPT, and light-dark cycling tests, the perovskite films were modified with OAI (5 mg in 1 mL isopropanol) through spin-coating at 4000 rpm for 30 seconds and subsequently annealed at 150°C for 10 minutes to form a 2D perovskite surface layer. The Spiro-OMeTAD/PTAA mixed solution (90/7 mg/mL in chlorobenzene) was supplemented with 35 μL Li-TFSI solution (260 mg/mL in acetonitrile), 30 μL tBP, and 30 μL FK209 Co(III) TFSI salt (375 mg/mL in acetonitrile), and deposited via spin-coating at 5000 rpm for 30 seconds.

Regarding thermal cycling and outdoor storage stability tests, the perovskite films remained unmodified with PEAI or other surface treatments. The PTAA solution (20 mg/mL in chlorobenzene) was mixed with 10 μL tBP and 2.6 μL Li-TFSI solution (260 mg/mL in acetonitrile), then spin-coated at 5000 rpm for 30 seconds.

Additionally, We have replaced all data with absolute PCE values instead of normalized data to more intuitively reflect the device stability (Figures 5A, 5C, 5E). Additionally, for Figure 5G, we have clearly labeled the initial PCE of the devices before 1000 h damp heat pre-exposure in the figure.

Besides these two main comments, there are several other issues that the authors need to address, as outlined below:

1# The authors reported their study on the anchoring ability of modifiers on oxide electrodes in Figure 1. It would be better to focus solely on SnO₂ in this section, as the inclusion of NiOx/SAM may cause confusion. Furthermore, the dissolution of DMF concerning the SAM is not inherently detrimental for pin perovskite cells, and the thickness of the SAM could potentially be reduced.

Our Response: Thank you very much for this comment. As shown in Figure R16 and Figure 1, we have removed all data and discussion regarding NiOx/SAM, retaining only the content focused on SnO₂ (Line 96, Page 5 in manuscript):

We focused on investigating the anchoring capability of modifiers on oxide electrodes. Common modifiers can be primarily categorized into ionic modifiers and molecular modifiers. For ionic modifiers, their binding to the oxide surface does not strictly form ionic bonds but rather primarily involves electrostatic adsorption, which constitutes a physical interaction. In the case of molecular modifiers, their binding to the oxide surface is mainly governed by relatively weak hydrogen-

bonding interactions. These interactions are relatively low in energy, making the modifier molecules prone to detachment (**Figure 1 A**). In contrast, DMF molecules in the liquid phase system contain an amide group (-CON-), which exhibits strong binding capability and can engage in robust interactions with most modifiers. Additionally, DMF is a highly polar solvent, where the modifier-DMF interaction is typically dominated by strong hydrogen bonding, supplemented by ion-dipole interactions, collectively enhancing the dissolution of the modifier in DMF. These two types of bonds represent relatively strong intermolecular forces, resulting in a comparatively higher adsorption energy. Consequently, during device fabrication, modifiers on the oxide surface are highly susceptible to dissolution and removal by perovskite precursor solvents, ultimately leading to the failure of the intended passivation effect.

To validate these hypotheses, we selected three representative modifiers to evaluate their anchoring effect on oxide electrodes. The structural formulas of these modifiers are presented in **Figure S1**. Specifically, inorganic salts (potassium chloride, KCl), organic salts (guanidine hydrochloride, $\text{CH}_5\text{N}_3\cdot\text{HCl}$), and ionic liquids (methylammonium acetate, MAAC) were used to modify the SnO_2 electrode commonly used in n-i-p type devices. We employed density functional theory (DFT) calculations to determine the binding energies of the aforementioned three modifiers with the electrodes, as well as their binding energies with the typical solvent DMF used in perovskite precursors (**Figure 1B**). As shown in **Figure 1C-E**, The adsorption energies between all modifiers and DMF were higher than those with the electrodes. We rinsed the modified electrodes with DMF and conducted X-ray photoelectron spectroscopy (XPS) analysis. After rinsing the KCl-modified SnO_2 electrode with DMF, the intensity of the K 2p peak decreased (**Figure 1F**), and the ratio of the K 2p_{3/2} peak area to that of the lattice O atoms dropped from 19.0% to 7.7% (**Figure S2 A and Table S1**), indicating desorption of KCl from the SnO_2 electrode surface. For the $\text{CH}_5\text{N}_3\cdot\text{HCl}$ -modified SnO_2 electrode, after rinsing with DMF, the intensity of the N 1s peak decreased (**Figure 1G**), and the ratio of the N 1s peak area to that of the lattice O atoms dropped from 22.4% to 15.9% (**Figure S2 B and Table S2**), representing desorption of $\text{CH}_5\text{N}_3\cdot\text{HCl}$ from the SnO_2 electrode surface. In **Figure 1H**, after rinsing the MAAC-modified SnO_2 electrode with DMF, the intensity of the C 1s peak decreased, and the ratio of the C 1s peak area to the lattice O atoms decreased from 18.1% to 12.3% (**Figure S2 C and Table S3**), indicating desorption of MAAC from the SnO_2 electrode surface¹⁷. These experimental results are consistent with the theoretical calculation results. Additionally, contact angle test results also confirmed this conclusion (**Figure S3**).

Figure R16. Study on the anchoring ability of modifiers on oxide electrodes (The modified Fig. 1).

2# In Figure 3F, the perovskite exhibits a strong PbI₂ peak even after modification; however, there appears to be no PbI₂ visible in the SEM image shown in Figure 3G.

Our Response: Thank you very much for this comment. In the fabrication of perovskite films via the full air process, real-time experimental conditions significantly impact the crystallinity quality of the films. Therefore, we sincerely apologize for the suboptimal performance of the previously prepared batch of perovskite films due to experimental condition limitations. We have repeated the experiments and prepared a new batch of higher-quality films, followed by additional XRD

characterization. As shown in Figures R17 and 3F, the observed trends remain consistent with previous findings. After co-modification of the substrate with DTPA and Zol, the crystallinity of the deposited perovskite films showed remarkable improvement.

Figure R17. XRD patterns of FAPbI₃ films grown on different SnO₂ electrodes before and after modification.

3# According to the device fabrication section, the authors used the two-step spin-coating or anti-solvent method to fabricate perovskite films. These two processes are commonly conducted in a glove box, except for the annealing step of the perovskite films prepared via the two-step spin-coating method. Did the authors use the anti-solvent method to fabricate perovskite solar cells in ambient air? If so, please specify this in the Device Fabrication section. Additionally, where was the spiro-OMeTAD layer deposited? If it was in a glove box, please mention it as well.

Our Response: Thank you very much for this comment. We have incorporated all your suggested experimental details into the manuscript (Line 394-432, Page 20-21). The specific additions are reproduced below:

Device fabrication:

SnO₂ Substrate Preparation: The FTO glass was successively cleaned in detergent, deionized water, and ethanol for 30 min by an ultrasonic cleaner. The clean FTO was treated with an ultraviolet ozone machine for 30 min. The SnO₂ solution was mixed with water at a volume ratio of 1:3 and spin-coated on the FTO at 3000 rpm for 30 seconds, followed by annealing at 150 °C for 30 min in air.

PSCs Fabrication: The whole process of the PSCs fabrication was carried out in an ambient air environment (room temperature and 20-50% relative humidity). Perovskite films were fabricated by either one-step spin-coating or two-step spin-coating. For surface passivation, 4 mg/mL PEAI in isopropanol was spin-coated on the perovskite film at 5000 rpm for 30 s without further

treatment. The hole transfer material (HTM) solution was prepared by mixing 90 mg Spiro-OMeTAD, 35 μl bis(trifluoromethane) sulfonimide lithium salt (Li-TFSI) solution (260 mg Li-TFSI in 1 ml acetonitrile), 30 μl 4-tertbutylpyridine (tBP), and 30 μl of FK209 Co(III) TFSI salt (375 mg/mL in acetonitrile). Then, the HTM solution was deposited on the top at 3000 rpm for 30 s in air without annealing. Finally, 80 nm Au was deposited to form an electrode by thermal evaporation under 5×10^{-4} Pa. Before the J-V test, a commercial anti-reflection layer (Shengyu Tech.) is carefully stuck on the glass side for better light transmittance.

(1) The perovskite films which were fabricated by the two-step solution process:

Firstly, 692 mg PbI_2 was dissolved in 1 ml DMF: DMSO (v: v=9:1), spin-coated on the SnO_2 electrode at 1500 rpm for 30 seconds, and annealed at 70 °C for 1 minute. After cooling down to room temperature, the organic solution (92.5mg FAI, 6.5mg MAI, and 9.5mg MAI in 1mL IPA, stirred for more than 2 h) was spin-coated onto the PbI_2 film at 2500 rpm for 3 s, 1350 rpm for 15 s and 1700 rpm for 12 s. The film was annealed at 150 °C for 15 min. All spin-coating and annealing processes were performed under ambient air conditions (room temperature and 20-50% R.H.).

(2) The perovskite films which were fabricated by the one-step solution process³⁸:

The perovskite precursor solution was prepared by dissolving 810 mg of black-phase FAPbI_3 , 35 mol% of MAI, and 1 mol% of acetylcholine chloride in DMF and DMSO (8:1 v/v) at 60 °C. The perovskite layers were then spin-coated at 1,000 rpm for 10 s and 5,000 rpm for 15 s, and 1 mL of ethyl ether was dripped onto the electrode during spin-coating. The perovskite layers were annealed at 120 °C for 40 min, and the electrode was cooled. All spin-coating and annealing processes were performed under ambient air conditions (room temperature and 20-50% R.H.).

4# What solvent was used in Fig. S2? The results show that the contact angle increased after rinsing with DMF. If deionized water was used to measure the contact angle, please specify this in the manuscript or in the captions of Fig. S2 and Fig. S12.

Our Response: Thank you very much for this comment. As you pointed out, the discussion and analysis of contact angle measurements in the original manuscript were indeed insufficient. We have now supplemented detailed descriptions of the contact angle testing procedures and analysis in the Supporting Information (Page 4, Figure S3; Page 17, Figure S15). The specific additions are reproduced below:

The contact angle measurements were performed using ultrapure water droplets as the testing liquid. Figure S3 A-C present contact angle measurements for different SnO_2 substrates: (A) SnO_2/KCl , (B) SnO_2/Gua , and (C) SnO_2/MAAc . All substrates initially exhibited excellent wettability. These substrates were then subjected to extensive DMF washing using the following protocol: 1000 μL of DMF was deposited onto each SnO_2 substrate, allowed to stand for approximately 15 seconds for complete wetting, followed by spin-coating at 3000 rpm for 30 seconds, and finally heated at 150°C for 5 minutes to remove residual solvent. Figure S3 D-F show the corresponding contact angles after intensive DMF washing for (D) SnO_2/KCl , (E) SnO_2/Gua , and (F) SnO_2/MAAc . Notably, all substrates demonstrated significantly increased contact angles

post-washing. This phenomenon can be primarily attributed to: The hydrophilic/polar nature of the surface modifiers - their removal via DMF washing reduced the polar component of surface energy. Consequently, the decreased surface modifier content diminished the substrates' hydrophilicity, manifesting as increased contact angles (Line 35 in SI).

Contact angle measurements were performed using ultrapure water droplets as the test liquid. Figure S15 A-D shows the contact angles of different SnO₂ substrates: (A) pure SnO₂, (B) SnO₂-DTPA, (C) SnO₂/Zol, and (D) SnO₂-DTPA/Zol. All modified SnO₂ substrates demonstrated better wettability than pure SnO₂. The substrates were washed with large amounts of DMF. Specifically, 1000 μL DMF was dropped onto different SnO₂ substrates, allowed to stand for about 15 seconds for complete wetting, then spin-coated at 3000 rpm for 30 seconds, and heated at 150°C for 5 minutes to remove the solvent. Figure S15 E-H shows the corresponding contact angles of SnO₂ substrates after extensive DMF washing: (E) pure SnO₂, (F) SnO₂-DTPA, (G) SnO₂/Zol, and (H) SnO₂-DTPA/Zol. After extensive DMF washing: The contact angle of pure SnO₂ substrate showed almost no change, proving that DMF washing treatment has little effect on the wettability of SnO₂ itself. The contact angles of SnO₂-DTPA and SnO₂/Zol substrates increased significantly, indicating partial desorption of DTPA and Zol molecules from the SnO₂ surface. The contact angle of the SnO₂-DTPA/Zol sample showed almost no change and still exhibited good wettability. This proves that co-modification with DTPA and Zol can ensure that SnO₂ does not fail due to desorption under the load of upper film solvents (Line 152 in SI).

5# The authors focus on PSCs fully fabricated in ambient air. I suggest adding the following article (*Adv. Energy Mater.* 2024, 14, 2400416) to Table S8.

Our Response: Thank you very much for your suggestion. We have incorporated the recommended references (*Adv. Energy Mater.* 2024, 14, 2400416) into Table S8 (Table R6).

Table R6. Summary of PCEs of high-efficiency (PCE > 24.5%) devices prepared in a full-air environment.

Perovskite	Relative Humidity (%)	Champion PCE (%)	Device structure	References
FAPbI ₃	20-40	25.52 (certificated 25.49)	n-i-p	This Work
FAPbI ₃	20	24.7	n-i-p	Y. Zou, Science , 2024 ¹

FAPbI ₃	30-85	25.74 (certificated 25.43)	n-i-p	Y. Yang, Adv. Energy Mater. 2024 ²
CS _{0.01} (FA _{0.97} MA _{0.03}) _{0.99} Pb(I _{0.97} Br _{0.03}) ₃	35-50	24.72	p-i-n	H. Meng, Nature Energy , 2024 ³

6# As shown in Fig. S21, the intensity of the XRD peak at 12.7° (PbI₂) decreased after the aging experiments. However, according to the authors' statement, the perovskite films decomposed during the UV light aging experiments, which should have resulted in an increased PbI₂ peak intensity.

Our Response: Thank you very much for this comment. We have improved the labeling in Figure S26 (The original Figure S21) and revised the corresponding analysis (Line 235-240, Page 29 in SI). After UV aging tests, the SnO₂-DTPA/Zol-based samples maintained a strong perovskite phase with only minimal δ -phase formation. The SnO₂/Zol-based samples exhibited more substantial δ -phase content while retaining partial perovskite phase stability. In contrast, the SnO₂-DTPA-based samples showed nearly complete perovskite phase degradation but preserved significant δ -phase components. The pure SnO₂-based samples demonstrated near-total signal loss with only trace δ -phase presence, suggesting almost complete decomposition into an amorphous state (Figure R18).

Figure R18. The XRD patterns of UV light aging experiments for perovskite films grown on different SnO₂ substrates. (254nm UV, T_{ambient}=35°C, RH=85%).

7# “2,5-dihydroxyterephthalic acid (H4DOBDC, C₈H₆O₆, >98.0%, TCI), nickel acetate tetrahydrate (Ni(CH₃CO₂)₂·4H₂O, 99.0%, Aladdin), zinc acetate dihydrate (Zn(CH₃CO₂)₂·2H₂O, 99.0%, Aladdin)”—Did authors use these three materials in this work?

Additionally, please provide the source information for diethylenetriaminepentaacetic acid (DTPA), zoledronic acid (Zol), SnO₂, Li-TFSI, and FK209 Co(III) TFSI salt used in this work.

Our Response: Thank you very much for this comment. We did not use these three materials in this work. Sorry for this mistake. We have revised the relevant descriptions in the Experimental section and supplemented the chemical information (Line 383-392, Page 19-20 in manuscript):

All materials have not been further purified since purchase. FTO/glass and Methylammonium iodide (MAI, >99.99%) were obtained from Advanced Election Technology. SnO₂ colloid precursor (tin(IV) oxide, Lead (II) iodide (PbI₂, 99.999%), Phenethylammonium iodide (PEAI, 99.5%), Spiro-OMeTAD (99.5%), PTAA (MOS) were purchased from Xi'an Polymer Light Technology Corp. Formamidinium iodide (FAI, >99.99%) and Methylammonium chloride (MACl, >99.99%) were purchased from Greatcell Solar Materials Pty Ltd. N, N-dimethylformamide (DMF), dimethyl sulfoxide (DMSO), isopropanol (IPA) and chlorobenzene (CB), 4-tert-butylpyridine (tBP), acetonitrile (Ace), bis(trifluoromethylsulfonyl)-imide lithium salt (Li-TFSI; 99.95%), were obtained from Sigma-Aldrich. FK209 Co(III) TFSI were purchased from Luminescence Technology Corp. Diethylenetriaminepentaacetic acid (DTPA, 99%), Zoledronic acid (Zol, 99%), were purchased from Aladdin.

8# Some revisions are needed. On page 4, “normal (nip) and inverted (pin) devices” should be corrected to “normal (n-i-p) and inverted (p-i-n) devices”. Please provide the full name of “MPP tests” on page 4 and “Voc” on page 10. Additionally, the full name of “Voc” should be “open-circuit voltage”, rather than “voltage open circuit” (line 219, page 12).

Our Response: We sincerely appreciate your suggestions. We have carefully addressed all the details you raised in our revisions (Line 79, Page 4; Line 89, Page 4; Line 195, Page 10 and Line 254, Page 13).

9# I disagree with the authors' statement on page 12: “The PL peak intensity of the SnO₂-DTPA/Zol perovskite film was the lowest, indicating that the modified ETL/perovskite interface had a higher crystallinity, lower defect density, and reduced charge recombination.” The PL results alone cannot confirm higher crystallinity at the modified ETL/perovskite interface.

Our Response: Thank you very much for this comment. We have revised the PL analysis (Line 273, Page 14 of the manuscript):

Compared to the pristine SnO₂/PVK sample, the SnO₂-DTPA/Zol/PVK sample exhibits significantly reduced photoluminescence peak intensity, demonstrating that the co-modification enhances the electron extraction capability of the SnO₂ ETL, thereby facilitating more efficient charge transfer from the perovskite to the ETL.

10# What hole transport material was used for the thermal stability tests? As far as I know, the spiro-OMeTAD layer is unstable at 85°C, which leads to device degradation. If the authors used a different hole transport material to evaluate PSC thermal stability, please provide more details about the fabrication process.

Our Response: Thank you very much for this comment. In light of the inherent instability issues associated with Spiro-OMeTAD as a hole transport layer, we incorporated PTAA into the Spiro-OMeTAD formulation for stability test samples due to PTAA's superior thermal stability. The detailed experimental protocol has been added to Line 422-442, Page 21-22 of the manuscript:

Under heating conditions, PEAI could be gradually converted into the PEA_2PbI_4 phase, leading to reduced device operational lifetime³⁹. Consequently, we removed the PEAI passivation layer for all thermal-related stability tests. Additionally, given the intrinsic instability of Spiro-OMeTAD as a HTL⁴⁰⁻⁴², we adopted a Spiro-OMeTAD/PTAA hybrid HTL for most stability evaluations, while employing pure PTAA HTL for a limited number of tests to ensure enhanced stability.

Specifically, for thermal stability testing, the perovskite films received no surface modification with PEAI or any other agent. The Spiro-OMeTAD/PTAA mixed solution (90/15 mg/mL in chlorobenzene) was supplemented with 35 μL Li-TFSI solution (260 mg/mL in acetonitrile), 30 μL tBP, and 30 μL FK209 Co(III) TFSI salt (375 mg/mL in acetonitrile), followed by spin-coating deposition at 5000 rpm for 30 seconds.

For storage, MPPT, and light-dark cycling tests, the perovskite films were modified with OAI (5 mg in 1 mL isopropanol) through spin-coating at 4000 rpm for 30 seconds and subsequently annealed at 150°C for 10 minutes to form a 2D perovskite surface layer. The Spiro-OMeTAD/PTAA mixed solution (90/7 mg/mL in chlorobenzene) was supplemented with 35 μL Li-TFSI solution (260 mg/mL in acetonitrile), 30 μL tBP, and 30 μL FK209 Co(III) TFSI salt (375 mg/mL in acetonitrile), and deposited via spin-coating at 5000 rpm for 30 seconds.

Regarding thermal cycling and outdoor storage stability tests, the perovskite films remained unmodified with PEAI or other surface treatments. The PTAA solution (20 mg/mL in chlorobenzene) was mixed with 10 μL tBP and 2.6 μL Li-TFSI solution (260 mg/mL in acetonitrile), then spin-coated at 5000 rpm for 30 seconds.

11# As shown in Fig. S11, an energy barrier of 40 mV was observed at the SnO₂/perovskite interface after DTPA and Zol modification, whereas no energy barrier was found at the interface of the control sample. Please revise the statement on page 10: "After being co-modified with DTPA and Zol, the valence band maximum (VBM) and conduction band minimum (CBM) of SnO₂ are shifted upwards, forming a more matched energy level alignment at the SnO₂/perovskite interface, which is expected to facilitate the extraction of electrons."

Our Response: Thank you very much for this comment. We have revised the relevant statement on Line 197-199 Page 10 of the manuscript as per your suggestion, and it now reads: "After co-modification with DTPA and Zol, both the valence band maximum (VBM) and conduction band minimum (CBM) of SnO₂ shift upward."

Additionally, to directly evaluate the impact of DTPA/Zol co-modification on interfacial charge transport, we conducted additional femtosecond transient absorption spectroscopy (fs-TAS) measurements. The results demonstrate that compared to pristine SnO₂/PVK, the SnO₂-DTPA/Zol/PVK sample exhibits significantly enhanced electron extraction rates (Line 199, Page 10 in manuscript and Figure R19, S14), indicating faster electron transfer from the perovskite to

the adjacent SnO₂ contact. We therefore conclude that the minor barrier observed in Original Figure S11 (now Figure S13) does not substantially impede charge carrier transport.

Figure R19. Femtosecond transient absorption spectroscopy (fs-TAS)

In the manuscript titled “Fully Chemical Interface Engineering for Statically and Dynamically Stable Perovskite Solar Cells,” Nie et al. introduced a chemical adsorption strategy to modify the interface between SnO₂ and perovskite, achieving a power conversion efficiency (PCE) of 25.52%. They also presented comprehensive stability studies demonstrating the durability of the devices. However, I believe the manuscript requires significant improvements before it can be reconsidered for publication in Nature Communications.

First, the reported efficiency has not been certified. It is now standard practice to certify the efficiency of perovskite solar cells, especially given the numerous measurement uncertainties associated with them. Moreover, it is worth noting that the in-house PCE claimed is significantly below the record PCE of 27% for perovskite solar cells.

Second, there seems to be a considerable amount of missing information in the experimental section. For instance, the authors claimed that the devices demonstrated stability at 85°C. This is questionable, as the use of the Spiro-MeOTAD hole transport layer in this device structure contradicts this claim; many studies have reported that doped Spiro-MeOTAD degrades at elevated temperatures. Additionally, in Figure 5, the authors presented normalized PCE instead of absolute PCE, which can be misleading. Devices with moderate performance may be able to improve their performance during aging tests, leading to an inaccurate representation.

Besides these two main comments, there are several other issues that the authors need to address, as outlined below:

1. The authors reported their study on the anchoring ability of modifiers on oxide electrodes in Figure 1. It would be better to focus solely on SnO₂ in this section, as the inclusion of NiO_x/SAM may cause confusion. Furthermore, the dissolution of DMF concerning the SAM is not inherently detrimental for pin perovskite cells, and the thickness of the SAM could potentially be reduced.
2. In Figure 3F, the perovskite exhibits a strong PbI₂ peak even after modification; however, there appears to be no PbI₂ visible in the SEM image shown in Figure 3G.
3. According to the *device fabrication* section, the authors used the two-step spin-coating or anti-solvent method to fabricate perovskite films. These two processes are commonly conducted in a glove box, except for the annealing step of the perovskite films prepared via the two-step spin-coating method. Did the authors use the anti-solvent method to fabricate perovskite solar cells in ambient air? If so, please specify this in the *Device Fabrication* section. Additionally, where was the spiro-OMeTAD layer deposited? If it was in a glove box, please mention it as

well.

4. What solvent was used in Fig. S2? The results show that the contact angle increased after rinsing with DMF. If deionized water was used to measure the contact angle, please specify this in the manuscript or in the captions of Fig. S2 and Fig. S12.
5. The authors focus on PSCs fully fabricated in ambient air. I suggest adding the following article (*Adv. Energy Mater.* 2024, 14, 2400416) to Table S8.
6. As shown in Fig. S21, the intensity of the XRD peak at 12.7° (PbI_2) decreased after the aging experiments. However, according to the authors' statement, the perovskite films decomposed during the UV light aging experiments, which should have resulted in an increased PbI_2 peak intensity.
7. "2,5-dihydroxyterephthalic acid (H_4DOBDC , $\text{C}_8\text{H}_6\text{O}_6$, >98.0%, TCI), nickel acetate tetrahydrate ($\text{Ni}(\text{CH}_3\text{CO}_2)_2 \cdot 4\text{H}_2\text{O}$, 99.0%, Aladdin), zinc acetate dihydrate ($\text{Zn}(\text{CH}_3\text{CO}_2)_2 \cdot 2\text{H}_2\text{O}$, 99.0%, Aladdin)"—Did authors use these three materials in this work? Additionally, please provide the source information for diethylenetriaminepentaacetic acid (DTPA), zoledronic acid (Zol), SnO_2 , Li-TFSI, and FK209 Co(III) TFSI salt used in this work.
8. Some revisions are needed. On page 4, "normal (nip) and inverted (pin) devices" should be corrected to "normal (n-i-p) and inverted (p-i-n) devices". Please provide the full name of "MPP tests" on page 4 and " V_{oc} " on page 10. Additionally, the full name of " V_{oc} " should be "open-circuit voltage", rather than "voltage open circuit" (line 219, page 12).
9. I disagree with the authors' statement on page 12: "The PL peak intensity of the SnO_2 -DTPA/Zol perovskite film was the lowest, indicating that the modified ETL/perovskite interface had a higher crystallinity, lower defect density, and reduced charge recombination." The PL results alone cannot confirm higher crystallinity at the modified ETL/perovskite interface.
10. What hole transport material was used for the thermal stability tests? As far as I know, the spiro-OMeTAD layer is unstable at 85°C , which leads to device degradation. If the authors used a different hole transport material to evaluate PSC thermal stability, please provide more details about the fabrication process.
11. As shown in Fig. S11, an energy barrier of 40 mV was observed at the SnO_2 /perovskite interface after DTPA and Zol modification, whereas no energy barrier was found at the interface of the control sample. Please revise the statement on page 10: "After being co-modified with DTPA and Zol, the valence band maximum (VBM) and conduction band minimum (CBM) of

SnO₂ are shifted upwards, forming a more matched energy level alignment at the SnO₂/perovskite interface, which is expected to facilitate the extraction of electrons.”